# Not-a-Bandit: Provably No-Regret Drafter Selection in Speculative Decoding for LLMs

**Hongyi Liu**[1][†][*]**, Jiaji Huang**[2]**, Zhen Jia**[2]**, Youngsuk Park**[2][†]**, Yu-Xiang Wang**[2,3][†]

[1]Rice University, [2]Amazon Web Services, [3]University of California, San Diego
hongyi.liu@rice.edu, {jiajihug, zhenj, pyoungsu}@amazon.com, yuxiangw@ucsd.edu

## Abstract

Speculative decoding is widely used in accelerating large language model (LLM) inference. In this work, we focus on the online draft model selection problem in speculative decoding. We design an algorithm that provably competes with the best draft model in hindsight for each query in terms of either the token acceptance probability or expected acceptance length. In particular, we show that we can accurately evaluate all draft models, instead of only the chosen model without incurring additional queries to the target model, which allows us to improve exponentially over the existing bandit-based approach as the number of draft models increases. Our approach is generically applicable with any speculative decoding methods (single draft, multi-drafts and draft-trees). Moreover, we design system-efficient versions of online learners and demonstrate that the overhead in computation and latency can be substantially reduced. We conduct extensive experiments on open-source LLMs and diverse datasets, demonstrating that our methods substantially outperform the state-of-the-art EAGLE3 and the BanditSpec baseline in a variety of domains where specialized domain-expert drafters are available, especially when long reasoning chains are required. Code available here.

## 1 Introduction

Speculative decoding (Chen et al., 2023; Sun et al., 2023; Li et al., 2025b) is widely adopted for accelerating large language models (LLMs). It uses a smaller *surrogate model*—referred to as a *draft model* or simply a *drafter*—to *predict* the sequence that a larger target model would generate. These predictions are then *verified* in parallel by the target model. When the drafters make correct guesses, a single expensive target pass is leveraged to produce multiple tokens, reducing per-token latency.

However, a single drafter may perform well on certain tasks but fail badly on others, leading to inconsistent quality of service and long-tail latency for certain queries. For example, a retrieval-based drafter works well when outputs closely match the input but falters elsewhere (Hou et al., 2025). Similarly, domain-specific drafters (e.g., for code, scientific writing, or summarization) excel in their own fields yet perform poorly outside them (Liu et al., 2023a; Kim et al., 2024; Yi et al., 2024). This raises a natural question: given access to multiple candidate drafters, how can we dynamically select the best one for each incoming query? Formally, we study the problem of **online drafter selection**:

*Given a pool of $N$ drafters, can we design an algorithm whose performance nearly matches that of adopting the best drafter in hindsight, for every user query?*

This problem was originally posed in MetaSD (Kim et al., 2024), who framed it as a *multi-armed bandit* task. Their method, together with the more recent BanditSpec (Hou et al., 2025), balances *exploration* (trying different drafters) and *exploitation* (using the empirically best drafter).

In this paper, we make the surprising observation that *exploration is unnecessary*. By carefully leveraging the structure of speculative decoding, we show that it is possible to efficiently compute feedback for *all* drafters—*not just the one selected*—without incurring additional calls to the target model. This transforms the problem from a bandit setting into a *full-information online learning*

---

[*]Work done during internship at AWS. [†]Correspondence to: Hongyi L., Youngsuk P., Yu-Xiang W.

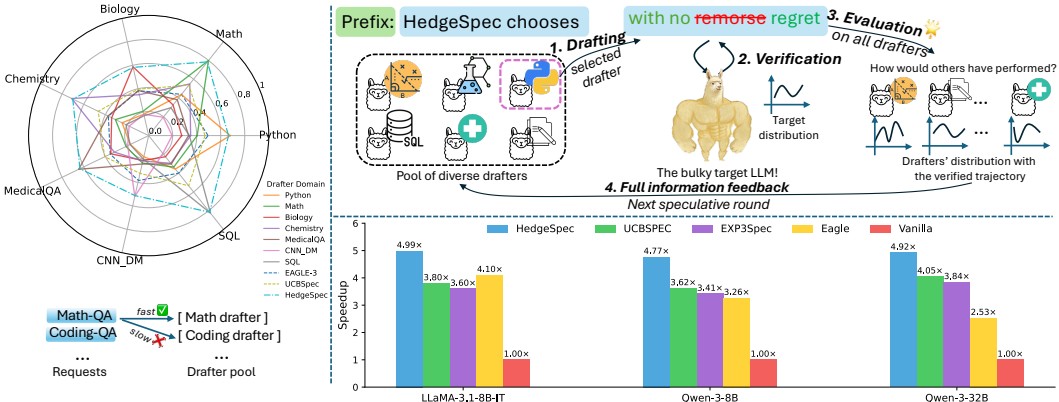

Figure 1: Overall workflow of HedgeSpec. **Left**: Acceptance rates across domains for Qwen-3-8B highlight each drafter's strong in-domain performance but steep decline outside its expertise, posing challenges for effective drafter selection. HedgeSpec achieves performance close to that of the best-performing expert. **Right-top**: Workflow of HedgeSpec. During the verification phase, a lightweight evaluation collects panoramic feedback from drafters by estimating the distribution gap between each drafter and the target. **Right-bottom**: Speedup ratio across baselines. HedgeSpec benefits from full-information feedback, achieving the highest speedup ratios.

problem. Building on this insight, we introduce HEDGESPEC, which uses algorithms such as Hedge (Littlestone & Warmuth, 1994; Vovk, 1995) or NormalHedge (Chaudhuri et al., 2009) to identify the best drafter *exponentially faster* in $N$ than bandit-based approaches.

Our key contributions are as follows:

- **Full-information framework for multi-drafter decoding:** We designed a novel method that accurately evaluates all drafters — not only the one that is chosen — with very low overhead, hence enabling full-information online drafter selection.

- **Theoretical guarantees:** We formulate and analyze the drafter selection problem under both acceptance probability and expected acceptance length objectives, establishing *no-regret guarantees*.

- **Empirical validation:** We conduct extensive experiments on LLaMA-3.1-8B-IT and Qwen-3-8B/32B reasoning model each with seven drafters, demonstrating that HEDGESPEC consistently outperforms EAGLE (up to 83.7% token/s gain in single domain and up to 46.1% on average) and bandit baselines (up to 49% MAT gain) in both acceptance rate and reduced per-token latency.

**Related work and novelty.** To the best of our knowledge, we are the first to show that full-information online learning is possible for drafter selection. Prior art (Hou et al., 2025) modeled the problem as a bandit problem. HedgeSpec could work with any given set of drafters on any speculative decoding method, thus making our contribution orthogonal to methodological innovation in speculative sampling and drafter curation (Sun et al., 2023; Cai et al., 2024; Li et al., 2024a; 2025b).

While we do not invent any new online learning algorithm, it is highly nontrivial to apply existing algorithms correctly to this problem. The core of our innovation is an off-policy estimator that returns the correct acceptance length *in expectation*, as well as to address a subtle *censoring issue* using learning from delayed feedback (Joulani et al., 2013).

On the system front, we designed a practical version of HedgeSpec that carefully balances the computational/latency cost of the evaluation and the statistical efficiency. As for the evaluation, we curated twenty-one drafters: seven dedicated to each target model, with every drafter specialized in a particular domain. We demonstrated that HedgeSpec with these drafters (a collection of specialists) outperforms the state-of-the-art EAGLE3 (a strong generalist) by a sizable margin.

## 2 PRELIMINARIES ON SPECULATIVE DECODING

### 2.1 SYMBOLS AND NOTATION

We denote the vocabulary set by $\mathcal{V}$, and a generated sequence by $x_{1:T}$, where each token $x_t \in \mathcal{V}$. The target language model defines a probability distribution $p(\cdot)$, while the draft (or surrogate) model defines $q(\cdot)$. A speculative decoding method $\mathcal{M}_q$ uses $q$ to generate $K$ speculative tokens $\hat{x}_{t+1:t+K}$ at chunk onset index $t_h$, which are then verified in parallel using $p$. We use $i$ to index multiple draft models $q_i$, $t$ for the global token position, $h$ for chunk indices, and $k \in [K]$ for token positions within a chunk. The token acceptance probability given that all previous tokens are correct is denoted by $\gamma_t[i]$. We adopt standard probability notation $p(\cdot)$, conditional probability $p(\cdot \mid \cdot)$, and conditional expectation $\mathbb{E}[\cdot \mid \cdot]$, with all probabilities assumed to be discrete.

### 2.2 BASIC SPECULATIVE DECODING METHOD

The basic speculative decoding algorithm proceeds as follows:

1. Generate a sequence of $K$ draft tokens $\hat{x}_{t+1:t+K}$ using a draft model $q$, i.e., $\hat{x}_{t+1:t+K} \sim q(x_{t+1:t+K}|x_{\leq t})$

2. For the target model, compute $p(x_{t+1}|x_{\leq t}), p(x_{t+2}|x_{\leq t+1}), ..., p(x_{t+K+1}|x_{\leq t+K})$ in parallel.

3. For $k = 1, 2, ..., K$, if $Z_k \sim \text{Uniform}([0,1])$, if $Z_k < \frac{p(\hat{x}_{t+k}|x_{\leq t+k-1})}{q(\hat{x}_{t+k}|x_{\leq t+k-1})}$, assign $x_{t+k} \leftarrow \hat{x}_{t+k}$ and continue; else draw $x_{t+k} \sim p_{\text{res}}(x) \propto \max\{0, p(x|x_{\leq t+k-1}) - q(x|x_{\leq t+k-1})\}$ and break.

4. If all $K$ tokens pass, assign $x_{t+K+1} \sim p(x_{t+K+1}|x_{\leq t+K})$.

**Theorem 1** (Leviathan et al., 2023, Theorem 3.5). *The samples generated from the speculative decoding algorithm with any draft model $q$ are drawn from the same distribution as $p$. The acceptance probability of the above algorithm is $\sum_{x \in \mathcal{V}} \min\{p(x), q(x)\} = 1 - \text{TV}(p, q)$ where $\text{TV}(p, q) = \frac{1}{2} \sum_{x \in \mathcal{V}} |p(x) - q(x)|$ denotes the total variation distance of two probability distributions.*

### 2.3 EAGLE AND DRAFT TREE

The EAGLE family is the most widely deployed speculative decoding models (Li et al., 2024a;b; 2025b). EAGLE-3 introduced multi-layer feature fusion with a training-time test mechanism. EAGLE-2 expands the draft tree with high-confidence tokens and prunes low-probability branches, improving efficiency by increasing the likelihood that more tokens are accepted per cycle. Specifically, instead of sampling from $q^{1:K}$ recursively like a language model, it generates a deterministic ([1]) draft-tree with Depth $K$ and branching factor $L$. From the root node ($x_{\leq t}$), EAGLE model $q$ generates $L$ children as possible choice of $x_{t+1}$. These are chosen to be the tokens with $L$ largest probabilities according to $q^1(\cdot|x_{\leq t})$. Then from each child, another $L$ descendants (subsequent tokens) are generated by sorting $q^2(\cdot|x_{\leq t+1})$. Different from speculative decoding, EAGLE drafters are not sensitive to the actual numerical values in $q$. Instead, only the relative ranking of the tokens matter. [2]

**Theorem 2.** *EAGLE with any draft model $q$ returns samples from the correct target distribution $p$. Moreover, the probability of accepting a token at step $k$ given all previously accepted tokens is $\sum_{v \in \text{Top}_L(q(\cdot|x_{\leq t+k-1}))} p(v|x_{\leq t+k-1})$. If dynamic pruning is used, then the candidate set $\text{Top}_L(q(\cdot|x_{\leq t+k-1}))$ should be replaced with the pruned set of descendants of $x_{\leq t+k-1}$.*

We defer the proof of Theorem 2 to Appendix A.1. All references to EAGLE in our paper refer to EAGLE-3 unless specified. We defer more related work discussion regarding speculative decoding, adaptive drafting and online algorithms in Appendix C.

### 2.4 PERFORMANCE METRICS FOR SPECULATIVE DRAFTER EVALUATION

We use two key metrics to evaluate a speculative decoding method $\mathcal{M}_{q,p}$: the *Expected Token Acceptance Probability (ETAP)* and the *Expected Acceptance Length (EAL)*. These capture complementary notions of per-token accuracy and end-to-end efficiency.

---

[1] It could also be randomized, but EAGLE–2 implementation focused on growing a deterministic greedy tree.

[2] The dependence on the values of $q$ becomes relevant when the *dynamic pruning* approach from EAGLE-2 is used, but it does not change the temperature-invariance of the original draft tree.

**Token Acceptance Probability (TAP).** At decoding step $t$, given a prefix $x_{<t}$, the drafter $q(\cdot)$ generates a speculative token $\hat{x}_t$. The token is *accepted* if it matches the target model $p(\cdot)$:

$$\gamma_t = \Pr_{\mathcal{M}_{q,p}} [\hat{x}_t = x_t \mid x_{<t}].$$

The **ETAP** is the expectation of this probability over a distribution of prefixes $\mathcal{D}$:

$$\text{ETAP}(\mathcal{M}_{q,p}, \mathcal{D}) = \mathbb{E}_{x_{<t} \sim \mathcal{D}}[\gamma_t].$$

ETAP measures the average per-token reliability of the drafter.

**Expected Acceptance Length (EAL).** Speculative decoding drafts $K$ tokens per chunk starting at index $t_h$, which are then verified by the target model. The *acceptance length* $\text{AcceptLength}(x_{\leq t_h})$ is the number of consecutive tokens accepted before the first mismatch. Even given a fixed prefix $x_{\leq t_h}$, AcceptLength remains a random variable because $\mathcal{M}_{q,p}$ often involves sampling. The **EAL** is:

$$\text{EAL}(\mathcal{M}_{q,p}, \mathcal{D}) = \mathbb{E}_{x_{\leq t_h} \sim \mathcal{D}} \big[ \mathbb{E}_{\mathcal{M}_{q,p}}[\text{AcceptLength}(x_{\leq t_h}) \mid x_{\leq t_h}] \big].$$

EAL reflects how many tokens are accepted per chunk and thus directly determines decoding efficiency: higher EAL means fewer calls to the target model, reducing time per token. We will later show how these two metrics are used to evaluate drafters.

**The Role of the Prefix Distribution.** Here prefix $x_{\leq t_h}$ refers not only to the initial prompt but also to the generated tokens that extend it up to step $t_h$. It is inherently random, determined jointly by the prompt, the stochastic rollout of the target model $p(\cdot)$, and the "tempo" of speculative decoding (i.e., where mismatches occur and chunks restart). The distribution $\mathcal{D}$ over prefixes provides a useful abstraction: it encapsulates all these factors into a single probabilistic view, allowing us to evaluate methods without conditioning on specific prompts or model trajectories.

## 3 HEDGESPEC: METHOD AND THEORY

We present HedgeSpec, a full-information online learning framework for drafter selection. By leveraging lightweight evaluation for panoramic feedback, HedgeSpec adaptively identifies most effective drafters, improving acceptance rates and end-to-end efficiency.

### 3.1 A NO-REGRET ONLINE LEARNING APPROACH TO DRAFT MODEL SELECTION

We are given $N$ draft models $q_1, ..., q_N$. At every time $t$, we decide which draft model to use to generate the next draft tokens. Our goal is to compete favorably with the best draft model that gives us either **(a)** the highest **acceptance probability** overall or **(b)** the highest **expected accepted length** per chunk. In this section, we will formulate this problem as a no-regret online learning problem, design appropriate loss functions that align with the two optimization objectives above, and develop algorithms that come with provable guarantees.

The performance guarantee is stated as a regret:

$$\text{Regret}_T = \sum_{t=1}^{T} f_t[i_t] - \min_{i^* \in [N]} \sum_{t=1}^{T} f_t[i^*]$$

where $f_t$ is the loss function at time $t$ determined by how well a draft model $q$ can perform at time $t$. For example, $f_t$ can measure the probability of not accepting tokens at time $t$ or how far the expected accepted length is from $K + 1$ where $K$ is the depth of the drafts.

### 3.2 FULL-INFORMATION-EVALUATION: HEDGESPEC IS NOT A BANDIT

Our main algorithmic idea is to add an evaluation phase between token verification and drafting. This evaluation phase was introduced in the BanditSpec paper. Differently, our work is motivated by that we can get feedback for all draft models, i.e., the full-information feedback, rather than only the draft model that we choose to play, i.e., the Bandit-feedback model. Adapting to different drafters changes generation speed, but thanks to Theorem 1 and 2, such adaptation does not change the distribution of

the output tokens. Moreover, as it will become clear, the additional evaluation on drafters does not require extra queries to the expensive target model, and can be carried out in parallel.

How does this evaluation phase work? A naive way is to roll out each drafter explicitly against the target, which is costly and infeasible. Instead, our key idea is that a single verified trajectory from the target model can serve as counterfactual evidence for evaluating all drafters. Concretely, after the chunk of new tokens $x_{t+1:t+k}$ are verified for $q_{i_t}$, we prefill them into $q_i$ for each of the other $i \in [N]/\{i_t\}$. This yields feedback from the trajectory, summarized by a sequence of feedback vectors:

$$\gamma_t[i] := \mathbb{P}_i[x_t \text{ is accepted}|x_{\leq t-1}] \text{ for } i \in [N], t \in [T]$$

For each method of speculative decoding, the acceptance probability is calculated differently. For example, for the standard speculative decoding with a single draft, $\gamma_{j,i} = 1 - \text{TV}[p(\cdot|x_{\leq t+j-1}), q_i(= \cdot|x_{\leq t+j-1})]$ by Theorem 1. For EAGLE's Greedy-Draft Tree approach $\gamma_{j,i}$ is the total probability of children nodes of parent $x_{\leq t+j-1}$ on the draft tree (constructed and pruned using $q_i$ in Theorem 2).

Once these feedbacks are collected, they form the basis for computing losses for each drafter, which will then be used to update the drafter-selection strategy. Crucially, our first result (Theorem 3) shows that the acceptance probabilities derived from the above verified trajectory are sufficient to construct an unbiased estimator of the acceptance length for any drafter.

**Theorem 3.** *Let $\mathcal{M}$ be a speculative decoding method and $K$ be the depth of its drafts. We write $x_{t+1:t+K} \sim p(\cdot|x_{\leq t})$ where $p$ refers to the target model. We denote $\gamma_k := \mathbb{P}_{\mathcal{M}}[x_{t+k} \text{ is accepted}|x_{t+1:t+k} \text{ are accepted}, x_{\leq t+k-1}]$. Also, define $\gamma_{K+1} = 0$ for notational convenience. The "one-step counterfactual" estimator:*

$$\widehat{AcceptLength}_{t,K}[\mathcal{M}] = \sum_{k=1}^{K+1} k(1 - \gamma_k) \prod_{j=1}^{k-1} \gamma_j$$

*satisfies that $\mathbb{E}_{\mathcal{M}}\left[\widehat{AcceptLength}_{t,K}[\mathcal{M}]\Big|x_{\leq t}\right] = \mathbb{E}_{\mathcal{M}}\left[\text{\# of accepted tokens}|x_{\leq t}\right].$*

The proof of Theorem 3 is included in Appendix A.2. This result is non-trivial because the probability $\gamma_k$ is not the probability of accepting the specific realized token $x_{t+k}$ given $x_{t+1:t+k-1}$ but rather the probability of accepting any token $\tilde{x}_{t+k} \sim p(\cdot|x_{t+1:t+k-1})$ even if $\tilde{x}_{t+k} \neq x_{t+k}$. We cannot compute the $\mathbb{E}\left[\text{\# of accepted tokens}|x_{\leq t}\right]$ directly because that would require access to all possible (combinatorial many) ways the target language model $p$ rolls out (see the illustration in Figure 2). The theorem proves that **as long as we have a single trajectory rolled out by the target model** (which we do since speculative decoding is lossless), we can counterfactually compute an unbiased estimator of the acceptance length for any alternative drafters.

**Variance of our estimator.** Observe that our length estimator has a bounded range $[1, K+1]$, thus by Popoviciu's inequality $\text{Var}\left[\widehat{AcceptLength}_{t,K}[\mathcal{M}] \Big| x_{\leq t}\right] \leq K^2/4$. In comparison, the estimator from BanditSpec (EXP3-Spec) is $O(NK^2)$, which grows with $N$.

**Connection to Experience Replay in Reinforcement Learning.** Our evaluation phase is similar to Experience Replay in the RL literature at a glance. However, we do

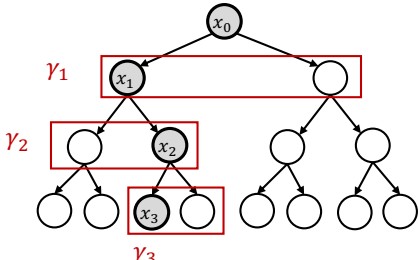

Figure 2: Illustration of the conditional acceptance probability $\gamma_k$ for the EAGLE model. Observe that $\gamma_k$ is not the probability of accepting $x_k$ only, but rather the total probability of accepting a token at level $x_k$. Also note that for evaluating the estimator, we never need to compute the probabilities of the target model for the possible alternative trajectories that the draft models generate.

not have the additional challenges in solving the distribution-shift induced by the policy used to collect the experience. For this reason, our "experience replay" for other draft models can be substantially more effective than that in RL.

### 3.3 HedgeSpec: Loss functions and Delayed Feedback

We have seen that a single verified trajectory suffices to estimate drafter's EAL. The remaining step is to formulate the online learning game: at each round $t = 1, 2, 3, ...$, nature generates a loss vector $f_t \in [0, 1]^N$, and the algorithm selects a drafter $i_t$ and incurs a loss of $f_t[i_t]$.

We just need to figure out how to instantiate it in our problem. There are several options. Each round of the game can be one token or one chunk of tokens. Also, we can choose the loss functions to optimize acceptance-length or acceptance probability. Pros and cons of these decisions are described in the Appendix B.3, but we find that the most natural setting is to directly optimize the acceptance length in a token-level game. The loss function is then chosen to be:

$$f_t[i] = 1 - \frac{1}{K+1} \sum_{k=1}^{K+1} k(1 - \gamma_{t+k-1}[i]) \prod_{j=1}^{k-1} \gamma_{t+j-1}[i].$$

By Theorem 3, the expected loss $\mathbb{E}[f_t[i]]$ measures how much room the $i_{th}$ drafter has to improve in its acceptance length from the maximal chunk length $K + 1$ if it starts at predicting token $t$. If we are to optimize acceptance probability instead, then we would choose $f_t[i] = 1 - \gamma_t[i]$.

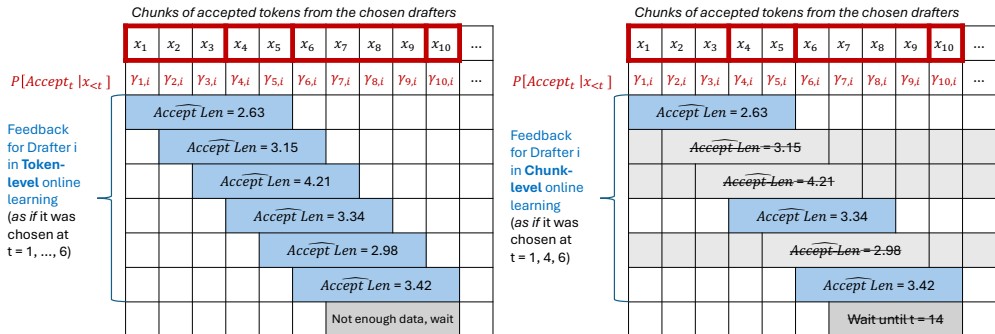

Figure 3: Illustration of the counterfactual feedback for Drafter $j$ (per-token or per-chunk) while the model keeps collecting new data. In this example, the generated token length $K = 5$. For the token-level learner, the feedback is delayed for $K$ steps, while for the chunk-level learner, the delay depends on the average number of chunks to collect $K$-tokens.

Readers with sharp eyes may notice a problem — the standard online learning game is not applicable! The accepted tokens are not observed immediately after token $t$, they are revealed in chunks after each speculative decoding chunk is completed. Moreover, unless the chosen drafter maxes out the acceptance length, there is not sufficient information to compute our estimator $\widehat{Accept\_Length}_{t,K}$. This is because alternative drafters may end up accepting more tokens than the drafter we chose. We refer to this problem as *censoring* issue. If we simply ignore the issue and compute the truncated version of the losses, we may end up getting stuck at a suboptimal solution (see the Appendix B.4 for an example). In other words, the learner may need to wait until multiple chunks of data to become available before getting the loss vectors for each time $t$ (see Figure 3 for an illustration).

---

for $t = 1, ..., T_{\text{token}}$ :

1. Nature chooses loss vector $f_t \in [0, 1]^N$

2. Player chooses Drafter $i_t$ for token $t$ and incurs a loss $f_t[i_t]$.

3. If $t$ is the end of a chunk, Player observes $f_{\leq t}$ (if optimizing acceptance prob) or observes $f_{\leq t-K}$ (if optimizing acceptance length).

---

Figure 4: Token-Level Game with Delay.

These issues can be modeled as "delayed feedback" (Weinberger & Ordentlich, 2002; Joulani et al., 2013; Van Der Hoeven & Cesa-Bianchi, 2022), which can be handled through a blackbox reduction to the standard online learning without delayed feedback with a mild increase in the regret as a function of the expected or maximum delay.

**Theorem 4.** *Assume $T_{Token} > 2K + 1$. Let $\mathcal{A}$ be ([Joulani et al., 2013](#), Algorithm 1) instantiated with Hedge or NormalHedge as BASE. Then $i_t$ chosen by $\mathcal{A}$ in the game defined in Figure [4](#) satisfies that:*

*1. When we optimize acceptance probability:*

$$\frac{1}{T_{token}} \sum_{t=1}^{T_{token}} \mathbb{P}_{\mathcal{A}}[x_t \text{ is accepted}] \geq \max_{i^* \in [N]} \left\{ \frac{1}{T_{token}} \sum_{t=1}^{T_{token}} \mathbb{P}_{i^*}[x_t \text{ is accepted}] \right\} - O\left(\sqrt{\frac{K \log N}{T_{token}}}\right).$$

*2. When we optimize accepted length:*

$$\frac{1}{T_{token}} \sum_{t=1}^{T_{token}} \mathbb{E}_{\mathcal{A}}[AcceptLength_{t,K}[i_t]] \geq \max_{i^* \in [N]} \left\{ \frac{1}{T_{token}} \sum_{t=1}^{T_{token}} \mathbb{E}[AcceptLength_{t,K}[i^*]] \right\} - O\left(\sqrt{\frac{(K+1)^3 \log N}{T_{token}}}\right).$$

The proof (rigorously written in Appendix [A.3](#)) applies the reduction to the regret bounds without delay in Theorem 1 of ([Joulani et al., 2013](#)) by noting that the max delay is $2K$. Then the result follows by taking expectation (over the distribution induced by the target model) on both sides and applying the definition of $\gamma_t[i]$ (or the unbiasedness of the length estimator in Theorem [3](#)).

The result says that the choices made by the online learner is nearly as good as the optimal choice in terms of either the average acceptance probability, or the expected accepted length.

### 3.4 Practical algorithms and extensions

In this section, we briefly describe the online learning algorithms that we chose to implement for the experiments. More detailed algorithms blocks can be found in the cited references therein. In our paper, we adopt NormalHedge ([Chaudhuri et al., 2009](#)) as our base algorithm (details in Figure [7](#)). For discussion regarding more hedging variants, please refer to [D.6](#) for detailed discussion.

**Handling delays.** We use the algorithms in ([Joulani et al., 2013](#)) for handling delay. In particular, the theoretical results above can be obtained by using ([Joulani et al., 2013](#), Algorithm 1), which handles bounded but non-constant delay (([Weinberger & Ordentlich, 2002](#)) requires constant delay). Moreover, since our problem is not really adversarial, but rather a Markov process induced by the target LLM, we found that ([Joulani et al., 2013](#), Algorithm 2) that operates in the "stochastic setting" works the best for us in practice. The algorithm is stated for the more general "partial monitoring" setting, but we are instantiating it in the full-information setting. Basically, the idea is to maintain a queue of the feedback and keep applying the next available actions generated by the base algorithm. In the iid setting, the regret is $\text{Regret}_T + O(\text{MaxDelay})$ with $\text{Regret}_T$ being the regret achieved by the base algorithm. We apply this algorithm as a heuristic (despite that the settings are iid) for the efficiency of learning in practice. More practical implementations heuristics can be found in [B.5](#).

## 4 Empirical evaluation

In this section, we provide comprehensive evaluations of HedgeSpec by conducting the following analysis: **1.** does HedgeSpec yield better **end-to-end performance**; **2.** an **in-depth analysis** of hedge based selection process. **3.** How does HedgeSpec perform **relative to offline based method**?

### 4.1 Experiment setup and baselines

We use Llama-3.1-8B-Instruct ([Dubey et al., 2024](#)), Qwen-3-8B and Qwen-3-32B ([Yang et al., 2025](#)) reasoning model as target models. In our main paper, all drafters in this paper are implemented with EAGLE-3, a widely used framework for speculative decoding. More experiments and discussion regarding the integration of HedgeSpec to broader speculative decoding framework can be found in Appendix [D.9](#). We adopt EAGLE's default generation configuration (see Section [D.1](#)). For hedge algorithm update, we use expected length to compute the loss and more detailed setup can be found in [B.5](#). We compare against the state-of-the-art drafter selection framework, BanditSpec ([Hou et al., 2025](#)), including both Exp3Spec and UCBSpec variants.

We report the Mean Number of Accepted Tokens (MAT) along with the wall-clock time required to complete each request which we use to compute token per second. All models are served using FP16 precision with a batch size of 1, following standard practices in latency-focused studies ([Fu et al., 2024](#); [He et al., 2023](#); [Cai et al., 2024](#)). Specifically, Token/s reflects the end-to-end latency

| Datasets | Python | | Math | | Biology | | Chemistry | | MedQA | | CNN_DM | | SQL | | | |
|---|---|---|---|---|---|---|---|---|---|---|---|---|---|---|---|---|
| Drafter domains | MAT | Token/s | MAT | Token/s | MAT | Token/s | MAT | Token/s | MAT | Token/s | MAT | Token/s | MAT | Token/s | Avg MAT | Avg Token/s |
| EAGLE | 6.48 | 87.37 | 5.88 | 76.35 | 5.95 | 71.20 | 5.28 | 71.28 | 4.96 | 66.48 | 5.31 | 67.26 | 5.99 | 80.41 | **5.69** | **74.34** |
| Python | **7.89** | **106.99** | 5.05 | 67.50 | 2.86 | 36.63 | 3.64 | 49.87 | 2.65 | 33.77 | 2.94 | 31.12 | 4.87 | 65.45 | 4.27 | 55.90 |
| Math | 4.52 | 62.77 | **8.03** | **106.43** | 3.07 | 39.46 | 4.15 | 55.74 | 3.02 | 42.22 | 3.32 | 44.79 | 4.36 | 60.16 | 4.35 | 58.79 |
| Biology | 3.29 | 35.34 | 3.91 | 47.46 | **7.27** | **96.10** | 4.35 | 56.42 | 4.42 | 56.44 | 3.17 | 36.19 | 2.72 | 28.78 | 4.16 | 50.96 |
| Chemistry | 3.80 | 51.84 | 6.46 | 86.28 | 4.45 | 59.93 | **7.39** | **96.28** | 3.82 | 50.42 | 3.09 | 32.56 | 3.76 | 46.90 | 4.68 | 60.60 |
| MedicalQA | 3.98 | 42.65 | 4.49 | 52.33 | 4.95 | 63.53 | 4.47 | 56.19 | **6.75** | **88.32** | 3.38 | 36.46 | 3.93 | 45.00 | 4.56 | 54.93 |
| CNN_DM | 2.07 | 26.85 | 2.89 | 38.79 | 3.02 | 40.55 | 2.99 | 39.21 | 3.11 | 42.22 | **6.20** | **77.19** | 2.14 | 28.12 | 3.20 | 41.85 |
| SQL | 3.71 | 44.16 | 3.66 | 46.47 | 2.64 | 36.34 | 2.96 | 31.84 | 2.62 | 34.15 | 2.90 | 37.94 | **8.49** | **114.52** | 3.85 | 49.35 |

Table 1: Statistics of the 7 curated drafters with Llama-3.1-8B-IT as the target (**Bold** = best). Each drafter performs strongly in-domain (diagonally) but suffers noticeable inefficiency when applied outside, and on average performs worse than the vanilla EAGLE model. Similar trends are observed in Qwen's drafters in Table 6. These drafters form a realistic evaluation pool for evaluating HedgeSpec, and we will see in Table 3, HedgeSpec orchestrates them to jointly accelerate the LLM inference.

during inference. These metrics are widely used (Hou et al., 2025; Li et al., 2025b) in speculative decoding and are positively correlated: longer accepted lengths typically lead to better throughput. All experiments are conducted on nodes with 8 NVIDIA A100 GPUs connected via NVLink. We defer jointly trained drafter discussion D.5 and hedging algorithm variants D.6 due to space limit.

## 4.2 Curating diverse drafters for large-scale evaluation

To thoroughly evaluate the effectiveness of our framework at scale, we build 21 drafters upon the official EAGLE-3 models (Li et al., 2025b; Tengyunw, 2025; Contributors, 2025) and finetune them on seven open-sourced datasets spanning multiple domains: Python (jtatman, 2025), Math (Toshniwal et al., 2024), Biology (Wesney, 2025), Chemistry (mlfoundations dev, 2025), MedicalQA (Chen et al., 2024), CNN_DM (Nallapati et al., 2016) and SQL (Meyer et al., 2024). The statistics of the resulting Llama and Qwen drafters are summarized in Table 1, 6 and 7. We observe that finetuning the generic EAGLE on a specific domain can greatly enhance its in-domain ability. We use SpecForge (Li et al., 2025a) as the training pipeline. In the meantime, no single model performs well across all domains, often exhibiting noticeable efficiency drops outside its specialization, and on average performs worse than the vanilla EAGLE. These drafters constitute a realistic experimental pool for evaluating our framework, and we will see that HedgeSpec orchestrates multiple drafters without prior knowledge to jointly accelerate the LLM inference process.

## 4.3 End-to-end efficiency analysis

We report the efficiency evaluation of HedgeSpec. We first conduct an overhead break down in HedgeSpec, followed by the end-to-end evaluation comparing with EAGLE and BanditSpec based drafting. Overall, HedgeSpec outperforms all other baselines with a significant margin consistently across every single domain, demonstrating its effectiveness.

### 4.3.1 Evaluation overhead breakdown

We analyze the evaluation overhead induced by incorporating global feedback for faster adaptation. This overhead comes from two major components: (i) prefilling drafters and (ii) computing losses and updating hedge weights. Table 2 reports the breakdown. Here, 'Llama' and 'EAGLE forward' denote forward passes through the target and drafter, respectively, while 'hedge update' includes both loss computation and NormalHedge weight updates. Evaluating a drafter

Table 2: Overhead comparison for drafter evaluation for Llama-3.1-8B-IT. Compared to an expensive target call, even a small boost in acceptance from HedgeSpec is enough to offset the cost of evaluating all drafters.

| Major Components | Llama Forward | EAGLE Forward | Hedge Update |
|---|---|---|---|
| Time/ms | 75.709 | 2.497 | 0.413 |

costs roughly 1/25 of a target forward, since EAGLE's drafter is a lightweight one-layer transformer compared to the 32-layer 8B Llama target. Under ideal assumptions, it implicates that if HedgeSpec secures one additional MAT, the gain offsets the cost of evaluating up to 25 drafters in sequence. In practice, the overhead is even smaller since drafter evaluations are independent and can be run in

| Datasets | Python | | Math | | Biology | | Chemistry | | MedQA | | CNN_DM | | SQL | | | |
|---|---|---|---|---|---|---|---|---|---|---|---|---|---|---|---|---|
| Methods | MAT | Token/s | MAT | Token/s | MAT | Token/s | MAT | Token/s | MAT | Token/s | MAT | Token/s | MAT | Token/s | Avg MAT | Avg Token/s |
| *LLaMA-3.1-8B-IT* | 1.00 | 17.03 | 1.00 | 18.87 | 1.00 | 18.44 | 1.00 | 18.25 | 1.00 | 18.57 | 1.00 | 17.86 | 1.00 | 17.86 | 1.00 | 18.13 |
| Eagle | 6.48 | 87.37 | 5.88 | 76.35 | 5.95 | 71.20 | 5.28 | 71.28 | 4.96 | 66.48 | 5.31 | 67.26 | 5.99 | 80.41 | 5.69 | 74.34 |
| UCBSpec | 5.44 | 74.39 | 5.75 | 77.57 | 5.22 | 70.27 | 5.11 | 71.79 | 4.46 | 61.14 | 3.94 | 50.47 | 5.71 | 76.58 | 5.09 | 68.89 |
| EXP3Spec | 5.16 | 69.29 | 5.59 | 74.21 | 4.93 | 67.70 | 4.93 | 64.25 | 4.25 | 58.13 | 3.81 | 49.70 | 5.37 | 73.29 | 4.86 | 65.22 |
| **HedgeSpec** | **7.69** | **99.58** | **7.69** | **98.63** | **7.18** | **93.78** | **7.10** | **89.65** | **6.47** | **77.26** | **5.88** | **70.68** | **8.06** | **103.31** | **7.15** | **90.41** |
| *Qwen-3-8B* | 1.00 | 14.55 | 1.00 | 14.58 | 1.00 | 14.64 | 1.00 | 14.52 | 1.00 | 14.59 | 1.00 | 14.49 | 1.00 | 14.65 | 1.00 | 14.57 |
| Eagle | 4.96 | 55.84 | 4.52 | 50.88 | 4.06 | 46.22 | 3.98 | 44.85 | 3.84 | 44.13 | 4.07 | 46.20 | 4.20 | 44.60 | 4.23 | 47.53 |
| UCBSpec | 4.75 | 54.80 | 5.30 | 61.28 | 4.16 | 47.29 | 4.73 | 54.37 | 4.25 | 48.96 | 3.58 | 41.07 | 5.28 | 61.28 | 4.58 | 52.72 |
| EXP3Spec | 4.54 | 52.77 | 5.10 | 59.06 | 4.03 | 46.50 | 4.46 | 50.66 | 4.07 | 45.02 | 3.39 | 37.53 | 4.97 | 56.74 | 4.37 | 49.75 |
| **HedgeSpec** | **6.32** | **68.52** | **7.27** | **79.55** | **5.66** | **62.08** | **6.61** | **73.18** | **6.08** | **65.82** | **5.10** | **54.97** | **7.52** | **81.94** | **6.37** | **69.44** |
| *Qwen-3-32B* | 1.00 | 8.13 | 1.00 | 7.96 | 1.00 | 8.43 | 1.00 | 8.33 | 1.00 | 8.20 | 1.00 | 8.06 | 1.00 | 8.55 | 1.00 | 8.21 |
| Eagle | 3.02 | 21.98 | 3.36 | 24.16 | 2.62 | 19.30 | 3.01 | 21.53 | 2.57 | 18.33 | 2.59 | 18.67 | 3.00 | 21.34 | 2.88 | 20.76 |
| UCBSpec | 4.34 | 31.42 | 5.24 | 38.69 | 4.25 | 31.73 | 4.76 | 35.58 | 4.19 | 30.99 | 3.64 | 26.93 | 5.07 | 37.36 | 4.50 | 33.24 |
| EXP3Spec | 4.22 | 30.33 | 5.13 | 37.42 | 4.10 | 30.31 | 4.54 | 33.46 | 3.96 | 28.76 | 3.52 | 25.32 | 4.88 | 35.29 | 4.33 | 31.55 |
| **HedgeSpec** | **5.82** | **38.44** | **6.96** | **45.14** | **5.93** | **39.13** | **6.50** | **42.86** | **5.92** | **38.60** | **5.19** | **33.38** | **7.16** | **45.30** | **6.21** | **40.41** |

Table 3: MAT (Mean Accepted Tokens) and Token/s (token generation rate) across datasets for each method. **Bold** indicates the best. Results of GSM8K and HumanEval and more other datasets, which is out of the training distribution, is in Table 12 showing the same trend. HedgeSpec consistently outperforms all baselines across domains. Its adaptive orchestration of expert drafters improves MAT and throughput, while full-information feedback delivers substantial gains over bandits, translating into real efficiency and highlighting HedgeSpec's effectiveness.

parallel. Thus, HedgeSpec is highly worthwhile: improved acceptance rates reduce costly target calls, outweighing the added cost of drafter evaluation.

### 4.3.2 HEDGESPEC ACHIEVES CONSISTENT END-TO-END GAINS ACROSS DOMAINS

We further present the end-to-end results of HedgeSpec against EAGLE and BanditSpec across seven testsets in Table 3. For GSM8K (Cobbe et al., 2021) and HumanEval (Chen et al., 2021) which are outside the training domains, similar trends hold and we defer details to Section D.8. The experiments presented here are conducted under the greedy inference setting of the target model. Results under non-greedy inference exhibit a similar trend, and a detailed discussion is deferred to Table D.4.

Overall, HedgeSpec consistently outperforms both EAGLE and bandit-based methods across all domains. Compared to EAGLE, its adaptive selection of speculative drafters enables effective orchestration, yielding faster responses. Notably, on SQL requests with Qwen, HedgeSpec improved MAT from 4.2 to 7.52 (an impressive 79% gain) and token/s from 44.6 to 81.94 (a 83.7% gain). Across all mixed queries, HedgeSpec achieved a strong 46.1% average improvement. While individual domain-specialized drafters perform worse on average than EAGLE, our results show that orchestrating them with HedgeSpec effectively leverages their strengths, leading to substantial efficiency gains.

HedgeSpec also surpasses bandit methods by a wide margin (up to 49% MAT gain and 41% token/s gain). Its advantage comes from the panoramic feedback: all drafters are evaluated, enabling faster convergence, higher acceptance, which finally leads to fewer target calls and better overall efficiency. In contrast, bandit learners adapt slowly because they only observe feedback from the chosen drafter, often converging to suboptimal choices and resulting in lower acceptance and throughput.

Interestingly, we also observe that bandit methods generally underperform EAGLE on LLaMA-3.1-8B-IT, whereas they outperform it on Qwen-3-8B. This is because the Qwen-3 series reasoning model tends to produce longer outputs than LLaMA-3.1-IT (i.e. 1.64x length in Math workload), providing more time for bandits to converge. The same phenomenon applies to HedgeSpec, where the longer reasoning chains in Qwen amplify its efficiency advantage, highlighting HedgeSpec's superiority in long-generation scenarios.

### 4.4 A DEEPDIVE INTO FULL-INFORMATION BENEFITS

We have seen HedgeSpec delivers better efficiency than EAGLE and bandit-based methods. In this section, we dive deeper into the benefit of full information on two aspects: (i) how their regret develops over time, and (ii) how each method scales with increasing number of drafters.

**HedgeSpec settles faster to near-zero (average) regret.** Figure 5 (a) shows cumulative regret on Llama-3.1 with python workload, measured by normalized EAL. Bandit methods accumulate regret quickly due to slow adaptation with only partial feedback, wasting exploration on weak drafters. In contrast, HedgeSpec rapidly converges to near-zero regret (detailed discussion in B.2) within a handful of steps, demonstrating the benefit of full information.

**HedgeSpec scales effectively with larger drafter pools.** We further plot MAT w/ number of candidate drafter increasing in Figure 5 (b). Scalability matters because larger pools raise the chance of including strong specialists, of course only works if drafter collaboration is effective. The upper and lower lines mark the best drafter and EAGLE's performance. Bandit methods deteriorate sharply with more drafters, as their regret grows quickly and exploration becomes much more costly. In contrast, HedgeSpec scales gracefully, being nearly unaffected because the global information helps to rapidly adapt on the best drafter, remaining effective even with large pool.

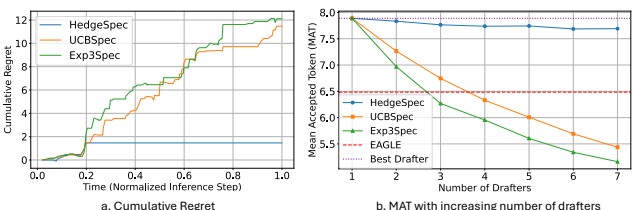

Figure 5: Cumulative regret and MAT vs. number of drafters. HedgeSpec quickly settles with near-zero regret, and can scale up with larger drafter pool. Qwen-3 results show similar trend in Appendix D.7, highlighting HedgeSpec's robustness.

### 4.5 OFFLINE ROUTER COMPARISON: HEDGESPEC'S ROBUSTNESS IN DISTRIBUTION SHIFT

In this section, we compare HedgeSpec with a static offline router. An offline router is a classifier trained on the request distribution and applied at serving time to dispatch queries to the best drafter. For fairness, we finetune a BERT (Devlin et al., 2019) classifier on the aggregated seven-domain dataset used to construct the drafters (hyperparameter in D.2). The classifier achieved 100% test accuracy across all domains, indicating that under closed-world assumptions, a lightweight classifier can reliably route requests to domain-specialized drafters.

|  | Llama-3.1-8B-IT | | Qwen-3-8B | |
|---|---|---|---|---|
| **Math** | MAT | Token/s | MAT | Token/s |
| Eagle | 5.83 | 78.05 | 4.63 | 54.49 |
| Static Router | 5.28 | 70.40 | 4.95 | 55.18 |
| **HedgeSpec** | **6.94** | **90.15** | **7.12** | **77.45** |
| **MedQA** | MAT | Token/s | MAT | Token/s |
| Eagle | 5.23 | 67.67 | 3.91 | 43.54 |
| Static Router | 3.03 | 40.83 | 2.49 | 28.18 |
| **HedgeSpec** | **5.56** | **70.38** | **6.04** | **65.99** |

Table 4: HedgeSpec vs. offline router under O.O.D (**Bold**=best). The router suffers and mis-routes requests to suboptimal drafters. HedgeSpec adapts through runtime feedback, and remains robust.

However, this approach critically assumes that all runtime queries remain in training distribution. In practice, cloud serving frequently encounters O.O.D prompts (Liu et al., 2023b; Chao et al., 2025; Cao et al., 2024). We came across a simple yet revealing case when attempting to elicit longer reasoning with the instruction: *'Please carefully read the question. After that, please generate two answers to validate it. Output the one you think works well.'* This minor prompt variation caused catastrophic failures: the classifier misrouted 98% of MedQA queries and 90% of Math queries. This was not even an adversarial attack but a natural prompt variation, underscoring the fragility of static routing. Such misrouting is costly as query dispatched to unsuitable drafter incurs long-tail overhead. As shown in Table 4, HedgeSpec remains robust under distribution shift, achieving up to 2.34× gains over the offline router. Its adaptive online learning leverages runtime feedback to identify the best drafter on-the-fly, offering three key advantages for real deployment: (i) no reliance on prior knowledge, (ii) resilience to O.O.D queries as long as experts remain useful, and (iii) adaptability when the prompt does not explicitly reveal the best drafter. Finally, we note that offline routing could complement HedgeSpec by providing a 'warm start' in the initial steps, which we leave for future investigation.

## 5 CONCLUSION

We present HedgeSpec, a full-information online drafter selection framework for speculative decoding. By leveraging the structure of speculative decoding, HedgeSpec evaluates all candidate drafters without incurring additional calls to the target model, enabling panoramic feedback. We establish theoretical guarantees under this setting and demonstrate substantial empirical gains, highlighting HedgeSpec's robustness and practical effectiveness in real-world LLM serving.

## ACKNOWLEDGMENTS

We thank Yida Wang from AWS for helpful discussion at an early stage of the project. We thank the anonymous reviewers and meta-reviewers for helpful feedback that led to improvements to the paper.

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

## A  PROOFS OF TECHNICAL RESULTS

### A.1  THEOREM 2 PROOF

*Proof.* EAGLE generates the draft tree greedily, thus the $L$ children at Node $x_{\leq t+k-1}$ is $\text{Top}_L(q(\cdot|x_{\leq t+k-1}))$. The next token is accepted when one of the children is sampled by the target model $p$.  □

### A.2  THEOREM 3 PROOF

*Proof of Theorem 3.* First note that the expectation is over the distribution of $x_{t+1:t+K}$ from the target distribution as well as the randomness of the speculative decoding method. In particular, $\gamma_k$ (even if the drafts are determinstic, e.g., in EAGLE3) is random because $x_{t+1:t+k-1}$ is random.

$$\mathbb{E}\left[\# \text{ of accepted tokens}|x_{\leq t}\right]$$

$$= \sum_{k=1}^{K+1} k\mathbb{P}\left[x_{t+1:t+k-1} \text{ is accepted}, x_{t+k} \text{ is not accepted}|x_{\leq t}\right]$$

$$= \sum_{k=1}^{K+1} k\mathbb{E}\left[\mathbb{P}\left[x_{t+k} \text{ is not accepted}|x_{t+1:t+k-1} \text{ is accepted}, x_{\leq t}\right]\Big|x_{\leq t}\right]$$

$$= \sum_{k=1}^{K+1} k\mathbb{E}\left[\mathbb{P}\left[x_{t+k} \text{ is not accepted}|x_{t+1:t+k-1} \text{ is accepted}, x_{\leq t+k-1}\right]\mathbb{P}\left[x_{t+1:t+k-1} \text{ is accepted}|x_{\leq t+k-1}\right]\Big|x_{\leq t}\right]$$

$$= \sum_{k=1}^{K+1} k\mathbb{E}\left[(1-\gamma_k)\mathbb{P}\left[x_{t+1:t+k-1} \text{ is accepted}|x_{\leq t+k-1}\right]\Big|x_{\leq t}\right] \tag{1}$$

Now by the Law of Iterated Expectation again

$$\mathbb{E}\left[(1-\gamma_k)\mathbb{P}\left[x_{t+1:t+k-1} \text{ is accepted}|x_{\leq t+k-1}\right]\Big|x_{\leq t}\right]$$

$$= \mathbb{E}\left[(1-\gamma_k)\mathbb{P}\left[x_{t+k-1} \text{ is accepted}|x_{t+1:t+k-2} \text{ is accepted}, x_{\leq t+k-2}\right]\mathbb{P}\left[x_{t+1:t+k-2} \text{ is accepted}|x_{\leq t+k-2}\right]\Big|x_{\leq t}\right]$$

$$= \mathbb{E}\left[(1-\gamma_k)\gamma_{k-1}\mathbb{P}\left[x_{t+1:t+k-2} \text{ is accepted}|x_{\leq t+k-2}\right]\Big|x_{\leq t}\right]$$

$$= \mathbb{E}\left[(1-\gamma_k)\gamma_{k-1}\gamma_{k-2}\mathbb{P}\left[x_{t+1:t+k-3} \text{ is accepted}|x_{\leq t+k-3}\right]\Big|x_{\leq t}\right]$$

$$= \quad \cdots$$

$$= \mathbb{E}\left[(1-\gamma_k)\gamma_{k-1}\gamma_{k-2}...\gamma_2\mathbb{P}\left[x_{t+1} \text{ is accepted}|x_{\leq t+1}\right]\Big|x_{\leq t}\right]$$

$$= \mathbb{E}\left[(1-\gamma_k)\gamma_{k-1}\gamma_{k-2}...\gamma_2\gamma_1\Big|x_{\leq t}\right]$$

In Line 4 onwards, we are repeatedly applying the same arguments from Line 1 - 3 which "peels off" one random token $j$ at a time and applying the definition of $\gamma_j$.

The proof is complete by substituting into (1).  □

### A.3  THEOREM 4 PROOF

*Proof of Theorem 4.* First, by the Hedge algorithm, the hypothetical online learning without delay enjoys a regret bound of $2\sqrt{T\log N}$ for any choice of $T$ and $N$.

By Theorem 1 of (Joulani et al., 2013), the blackbox reduction shows that when the delay is smaller than $2K$, there is an algorithm that gives a regret of

$$2(2K+1)\sqrt{\frac{2T}{2K+1}\log N}$$

as long as $T > (2K+1)$. This yields the following regret bounds

$$\frac{1}{T_{\text{token}}}\sum_{t=1}^{T_{\text{token}}}\gamma_t[i_t] \geq \max_{i^*\in[N]}\left\{\frac{1}{T_{\text{token}}}\sum_{t=1}^{T_{\text{token}}}\gamma_t[i^*]\right\} - O\left(\sqrt{\frac{K\log N}{T_{\text{token}}}}\right).$$

and

$$\frac{1}{T_{\text{token}}} \sum_{h=1}^{T_{\text{token}}} \frac{\widehat{\text{AcceptLength}}_{t,K}[i_t]}{K+1} \geq \max_{i^* \in [N]} \left\{ \frac{1}{T_{\text{token}}} \sum_{h=1}^{T_{\text{token}}} \frac{\widehat{\text{AcceptLength}}_{t,K}[i^*]}{K+1} \right\} - O(\sqrt{\frac{K \log N}{T_{\text{token}}}}).$$

for the two different loss functions respectively. It remains to take expectation over all random variables on both sides for each $i^* \in [N]$ separately. For the acceptance probability, notice that

$$
\begin{aligned}
\mathbb{E}[\gamma_t[i_t]] &= \mathbb{E}[\mathbb{P}_{i_t}[x_t \text{ is accepted } |x_{\leq t-1}]] \\
&= \mathbb{E}[\sum_{i \in [N]} \mathbb{P}_{\mathcal{A}}[i_t = i|x_{\leq t-1}]\mathbb{P}_i[x_t \text{ is accepted } |x_{\leq t-1}]] \\
&= \sum_{i \in [N]} \mathbb{E}[\mathbb{P}[i_t = i|x_{\leq t-1}]]\mathbb{E}[\mathbb{P}_i[x_t \text{ is accepted } |x_{\leq t-1}]] \\
&= \sum_{i \in [N]} \mathbb{P}_{\mathcal{A}}[i_t = i]\mathbb{P}_i[x_t \text{ is accepted }] = \mathbb{P}_{\mathcal{A}}[x_t \text{ is accepted}]
\end{aligned}
$$

where the third line uses the conditional independence of $i_t$ and $x_t$ given $x_{\leq t-1}$, which follows because the algorithm $\mathcal{A}$ decides on $i_t$ before $x_t$ and that $x_t$ is determined by the target model $p$ no matter which $i_t$ is chosen.

For the accepted length,

$$
\begin{aligned}
&\mathbb{E}\left[\widehat{\text{AcceptLength}}_{t,K}[i_t]\right] \\
=&\mathbb{E}\left[\mathbb{E}[\widehat{\text{AcceptLength}}_{t,K}[i_t]|i_t, x_{<t}]\right] \\
=&\mathbb{E}\left[\mathbb{E}[\text{AcceptLength}_{t,K}[i_t]|i_t, x_{<t}]\right] \\
=&\mathbb{E}\left[\sum_{i \in [N]} \mathbb{P}_{\mathcal{A}}[i = i_t|x_{<t}]\mathbb{E}[\text{AcceptLength}_{t,K}[i]|i, x_{<t}]\right] \\
=&\mathbb{E}\left[\text{AcceptLength}_{t,K}[i_t]\right],
\end{aligned}
$$

where the second identity follows from Theorem 3 and the third-identity uses the conditional independence of $i_t$ and $x_t$ given $x_{\leq t-1}$ as before. The claim is proven by multiplying both sides by $K+1$. $\qquad\square$

## B  FURTHER ALGORITHM AND IMPLEMENTATION DETAILS

### B.1  ONLINE LEARNING WITH DELAYED FEEDBACK IN STOCHASTIC SETTING

In this section, we clarify how Algorithm 2 of (Joulani et al., 2013), stated in the more general partial monitoring setting, can be instantiated for the full information setting.

This is the algorithm that we used in the experiment.

**Setting.**  There are $N$ experts (actions) $A = \{1, \ldots, N\}$. On each round $t = 1, 2, \ldots, T$ the environment associates a loss vector $\ell_t \in [0,1]^N$ to the experts, but $\ell_t$ may be *revealed after an arbitrary delay*. At time $t$ the learner receives a (possibly empty) set

$$H_t = \{(s, \ell_s) : \text{the full vector } \ell_s \text{ becomes available at time } t\}.$$

When making the round-$t$ decision, the learner can use all $\ell_s$ that have already been revealed, but not those still pending.

**Base learner.**  BASE is any standard *full-information* online algorithm (e.g., Hedge) designed for an immediate, delay-free stream of loss vectors. We assume BASE supports:

$$\text{PREDICT}() \quad \text{and} \quad \text{UPDATE}(\ell),$$

where PREDICT returns either a distribution over experts or a single expert index, and UPDATE feeds BASE one full loss vector $\ell \in [0, 1]^N$.

Specializing QPM-D to full information collapses the per-action queues into a *single FIFO queue* of unprocessed loss vectors, because any revealed $\ell_s$ is usable regardless of which expert was played.

---

**Algorithm 1:** Queued Full-Information with Delays (QFI-D)

**Data:** A FIFO queue $\mathcal{Q}$ of unprocessed loss vectors
**Input:** (Optional) horizon $T$ (not needed if the BASE is an *anytime* algorithm).

1   Initialize $\mathcal{Q} \leftarrow \emptyset$.
2   Initialize BASE; let $p \leftarrow$ BASE.PREDICT().
3   **for** $t = 1, 2, \ldots, T$ **do**
     // Predict phase: catch BASE up with all arrived feedback
4     **while** $\mathcal{Q} \neq \emptyset$ **do**
5        $\ell \leftarrow$ POPFRONT($\mathcal{Q}$)
6        BASE.UPDATE($\ell$)
7        $p \leftarrow$ BASE.PREDICT()
     // Make the real-world choice for round $t$
8     Play action $a_t$ according to $p$
     // e.g., sample from $p$ or take $\arg\max/\arg\min$ as appropriate
       for BASE
     // Update phase: record any feedback that arrives now
9     Observe the (possibly empty) set $H_t = \{(s, \ell_s)\}$ of loss vectors revealed at time $t$.
10    **foreach** $(s, \ell_s) \in H_t$ **do**
11       PUSHBACK($\mathcal{Q}, \ell_s$)

---

**Remarks.**

- This makes BASE experience a delay-free stream *in its own clock*: whenever a vector arrives, it is immediately fed to BASE before the next real prediction is made.

- To instantiate with **Hedge**, UPDATE applies the usual weight update $w_i \leftarrow w_i \exp(-\eta \, \ell_i)$ and PREDICT returns the normalized weights.

- If multiple loss vectors arrive at the same time, they are queued in arrival order and processed FIFO.

**Corollary 5** (Regret of QFI-D). *Assume the delay is bounded by $\tau_{\max}$ and the loss is bounded by $1$. Let* Base *be any full-information online learner analyzed in the same stochastic environment* without *delays, with expected regret* $\text{Regret}_T^{\text{Base}}$. *Then the expected regret of QFI-D satisfies*

$$\mathbb{E}[\text{Regret}_T] \leq \mathbb{E}\left[\text{Regret}_T^{\text{Base}}\right] + \tau_{\max}.$$

*Proof.* This is an instantiation of Theorem 6 of (Joulani et al., 2013). □

The implementation for HEDGESPEC with delayed feedback in the stochastic setting is particularly simple.

1. Keep two pointers $t_{\text{updated}}$ and $t$ where $t_{\text{updated}} \leq t$ and all necessary statistics $\gamma_{t > t_{\text{updated}}}$.

2. After every chunk, process every batch of available loss vectors by updating the weights of the BASE learner, before taking the next action, then update $t_{\text{updated}}$ to the last frontier.

Note that in the hypothetical token-level game without delay, there are several updates within each chunk, but due to the delay, none of those updates will actually occur. This means that the weights on the drafters in the delayed case will not be updated within each speculative decoding chunk is complete.

We could either sample independent sample from the drafter weights for each new token as the drafter roll out or stick to the same drafter. Both approaches enjoy the same regret guarantees in Corollary 5.

## B.2 FIRST ORDER AND SECOND ORDER REGRET BOUNDS

We observe that in the experiments (Figure 5 and Figure 8), HEDGESPEC appears to have a regret that stops growing after learning for a few iterations, instead of the $O(\sqrt{T})$ predicted by the worst-case bound in Theorem 4 and Corollary 5.

We believe this is due to the adaptivity of NormalHedge (Chaudhuri et al., 2009).

Cesa-Bianchi et al. (2007) established that when the learning rate is optimally tuned for Hedge, the method enjoys both first order (small loss) and second order (small variance) regret bounds:

$$\text{Regret} = O\left(\sqrt{\sum_{t=1}^{T} f_t[i^*] \log N}\right), \quad \text{and} \quad \text{Regret} = O\left(\sqrt{\sum_{t=1}^{T} \text{Var}_{i \sim p_t}[f_t[i]] \log N}\right).$$

In particular, if the best drafter $i^*$ has very small losses or after a while the learner's weights $p_t$ concentrates on a fixed drafter, then the regret bound will not grow with $T$. This is the case when there is a clear winner among all drafters.

NormalHedge was not proven to enjoy these strong adaptive regret bounds, though there was a conjecture that it does (Freund, 2016), and a modified version of normal hedge algorithm known as AdaNormalHedge (Luo & Schapire, 2015) which does enjoy first order regret bounds.

Our experiments seem to support the conjecture.

In practice, we find that NormalHedge often quickly converges to the optimal drafter, while still enjoy the worst-case $\sqrt{T}$-type regret when no clear winner exists.

## B.3 CHUNK-LEVEL VS TOKEN-LEVEL ONLINE LEARNING AND REGRET GUARANTEES

There is more than one way to set up the regret minimization game. In the main paper, we discuss the token-level online learning game. Here, we further discuss the token-level online game.

**Chunk-Level Games and Loss functions**  We can set it up as an online learning problem where each chunk is one round of the game. This choice is natural because the action to choose drafters is made for each chunk.

---

for $h = 1, ..., T_{\text{chunk}}$ :

    1. Nature chooses loss vector $f_h \in [0, 1]^N$.

    2. Player chooses Drafter $i_h$ and incurs loss $f_h[i_h]$.

    3. Player observes $f_h$.

---

Figure 6: Chunk-Level Game

It remains to design the loss functions. Let the token index right before the $h^{th}$ chunk be $t_h$ and the length of chunk returned by the chosen drafter be of length $k_h$. Let

$$\gamma_{h,j}[i] := \mathbb{P}_i[x_{t_h+j} \text{ is accepted} \mid x_{t_h+1:t_h+j-1} \text{ are accepted}, x_{<t_h+j}],$$

namely the probability that $j^{th}$ token of chunk $h$ from drafter $i$ is successfully validated given that the first $(j - 1)$ tokens are validated.

If we optimize the *average acceptance probability* within each chunk, we can use

$$f_h[i] = \frac{1}{k_h} \sum_{j=1}^{k_h} (1 - \gamma_{h,j}[i]).$$

If we optimize the expected chunk length, then we can apply the expression in Theorem 3 with $K = k_h$ to estimate the expected accept length of Drafter $i$. The resulting loss function is the

normalized distance from the max length.

$$f_h[i] = 1 - \frac{1}{k_h+1} \widehat{\text{AcceptLength}}_{h,k_h}[i] = 1 - \frac{1}{k_h+1} \left( \sum_{j=1}^{k_h+1} j(1 - \gamma_{h,j}[i]) \prod_{\ell \in [j-1]} \gamma_{h,\ell}[i] \right)$$

where $\gamma_{h,k+1}[i]$ is assigned to be 0 for notation convenience.

**Theorem 6.** *There is an algorithm $\mathcal{A}$ that chooses $i_h$ in the game in Figure 6, such that*

1. *When we optimize acceptance probability*

$$\frac{1}{T_{chunk}} \sum_{h=1}^{T_{chunk}} \frac{1}{k_h} \sum_{j=1}^{k_h} \gamma_{h,j}[i_h] \geq \max_{i^* \in [N]} \left\{ \frac{1}{T_{chunk}} \sum_{h=1}^{T_{chunk}} \frac{1}{k_h} \sum_{j=1}^{k_h} \gamma_{h,j}[i^*] \right\} - 2\sqrt{\frac{\log N}{T_{chunk}}}.$$

2. *When we optimize Accept Length*

$$\frac{1}{T_{chunk}} \sum_{h=1}^{T_{chunk}} \frac{\widehat{AcceptLength}_{h,k_h}[i_h]}{k_h+1} \geq \max_{i^* \in [N]} \left\{ \frac{1}{T_{chunk}} \sum_{h=1}^{T_{chunk}} \frac{\widehat{AcceptLength}_{h,k_h}[i^*]}{k_h+1} \right\} - 2\sqrt{\frac{\log N}{T_{chunk}}}.$$

The algorithm that achieves this includes Hedge and many of its variants. In the experiments, we test out both the original Hedge and more adaptive, and parameter-free variants of Hedge. Compared to BanditSpec, the regret improves exponentially in the number of drafters

Note that the actual accepted length for the chosen drafter model $i_h$ is $k_h$, but the expected value can be bigger or smaller than $k_h$. Other draft models will have an expected accepted length between 1 and $k_h + 1$. It is capped at $k_h + 1$ due to a *censoring effect*. The censoring effect may lead to an underestimation of the performance of alternative draft models. We will elaborate on these consequences of the censoring issue in the appendix.

Another issue of censoring is that it makes the regret guarantees in Theorem 6 somewhat difficult to interpret. In particular, it might be suprising to some readers that for any fixed $i$

$$\mathbb{E}[\widehat{AcceptLength}_{h,k_h}[i]] \neq \mathbb{E}[AcceptLength_{h,k_h}[i]].$$

in general. This is because $k_h$ is random, and when we condition on $k_h$, it changes the distribution of the tokens $x_{t_h+1:t_h+k_h}$ in this chunk, thus rendering Theorem 3 inapplicable. Similar issues arise for the expected value of $\gamma_{h,j}$.

### B.4 TIME-INHOMOGENEITY IN EAGLE MODELS AND CONSEQUENCE OF CENSORING.

EAGLE draft models are special in that they belong to a broader family of *time-inhomogenous* draft models. Let's denote EAGLE models by $q_{i,k}$ — the $i$th choice and the corresponding model used for $k$th relative position.

As an example, if Draft model $i = 1$ is chosen at time $t$ it rolls out with 2 tokens verified. Then the models being called are $q_{1,1}$ for the first token, and $q_{1,2}$ for the second. Support Draft model $i = 2$ was chosen instead, then it could've verified 4 tokens.

Naively, this can give us feedback for $q_{i,1}, q_{i,2}$ for $i = 2$ too, but will have to wait until the next chunk before we can get feedback for $q_{2,3}$ and $q_{2,4}$. What if the onset $t$ is now $t + 1$ instead?

The execution of the target model actually has provided us all the information needed to provide feedback for every $i \in [N]$ and $k \in [k]$

Why do we need to handle these complications? Let us inspect the following two examples that highlight the surprising failure mode if we do not handle these issues appropriately.

Evaluating EAGLE models off-policy is somewhat tricky because the best model is not fully captured by the acceptance probability.

**Example 7** ( "censoring" causes worse draft model win). *Draft model $q_{1,1}$ has acceptance probability 100% but $q_{1,2}$ is terrible, it has acceptance probability of only 0%. The expected verified length is always 2 for draft model $q_{1,:}$. If $q_{1,1}$ is played first, the chunk length will converge to 2 throughout.*

*Now let $q_{2,k} = 0.4$ for $k = 1, 2$ but $0.9$ for $k > 2$.*

*Average acceptance probability for draft model 1 is $0.5$ and $0.4$ for draft model 2 under the distribution of draft model 1. However, if we roll-out draft model 2 long enough, we get higher acceptance probability on average.*

*The expected acceptance length for the second draft model is $1 + 0.6/(1 - 0.6) = 2.2$, i.e. it is better than the first.*

*However, since draft model 1 is only generating two-token chunks, and we can only evaluate the first two steps of the draft model 2.*

*The expected length is now $0.4 + 0.6.4 \times 2 + 0.6 \times 0.6 \times 3 = 1.96 < 2$.*

This example illustrates that for drafters that are not invariant to relative index (e.g., EAGLE models), it is perhaps better to use token-level online learning (with delay) to avoid getting stuck at a suboptimal drafter.

In practice, we found that the censoring effect is not detrimental for EAGLE models. Chunk-level online learning works as well as token-level online learning with delay, and it incurs smaller system overhead.

### B.5   PRACTICAL HEURISTICS: "HYBRID LOSSES" "SKIPPING UPDATE"

As we now evaluate all drafters, even though this does not increase the total number of calls to the target model, it does *slightly* affect the amount of compute needed for evaluation and increase the latency in practice in the inference system.

We propose a technique called "skipping updates" which reduces the number of times the evaluation phase need to be called. The idea is that we simply grouping a couple of updates together and apply them once in a batch once in a while.

In practice, we observed that our hedging algorithm quickly reaches a stable performance plateau with near-zero regret, consistent with the first- and second-order regret bounds (see discussion in B.2). To further reduce update latency, we experimented with skipping updates by a fixed number of tokens and combining this with batched feedback, treating both as tunable hyperparameters. Empirically, after a short warm-up period (e.g., 6 rounds of full-information updates), using delayed batched feedback (e.g., 12 rounds per batch) together with skipped updates (e.g., 6 tokens per update) continued to yield strong performance for all the tested experiments. Overall, these results indicate that moderate delays in feedback and updates can maintain good mean accepted token length while providing a practical tradeoff between accuracy and efficiency.

The other trick that we proposed is "hybrid losses". This involves starting the learner by choosing the first few loss vectors, e.g., $f_1, f_2, f_3, ...$ to be based on acceptance probabilities, then switching to the acceptance-length loss, later. The reason is that the delay is generally higher for acceptance length.

For example, if $K = 8$ and after the first chunk, $4$ tokens are accepted. The acceptance rate loss would be computable after the first chunk. By the end of the second chunk, we can already update the learner by $4$ times before the decision for the second chunk is due.

The loss based on acceptance probability — even though not what we ultimately wanted to optimize — can be used as a surrogate loss and help mitigating the "cold-start" problem.

Both tricks were used in our experiments.

## C   RELATED WORK

### C.1   THE ADVANCE OF SPECULATIVE DECODING

Speculative decoding is a pivotal way for optimizing LLM inference latency. This technique was first introduced with chain-structured drafts, where the draft model generates a single sequence of tokens verified sequentially by the target model (Leviathan et al., 2023; Chen et al., 2023). Subsequent work generalized this into tree-structured drafts, organizing draft tokens as a connected tree to increase

acceptance opportunities (Chen et al., 2025; Miao et al., 2024; Cai et al., 2024; Du et al., 2024; Li et al., 2024b). Recent works extend speculative decoding to the multi-draft setting, where multiple candidate tokens are proposed in parallel at each step. Sun et al. (2023) casts draft selection into an optimal transport framework with efficient approximation schemes. Khisti et al. (2025) show that the optimal solution admits a canonical two-step decomposition and provide exact acceptance characterizations in the two-draft case. Hu et al. (2025) derive tractable methods to compute theoretical upper bounds on acceptance rates, demonstrating practical benefits of sampling strategies such as without-replacement. The most recent advances leverage diffusion LLMs (dLLM) (Wu et al., 2025) as drafters (Pan et al., 2025; Chen et al., 2026), further reducing drafting overhead and improving overall efficiency. **While prior work advances speculative decoding by improving how a single drafter generates candidates**—ranging from chain- to tree-based structures and multi-draft extensions—**our work addresses the orthogonal level of challenge in multi-drafter selection**, where diverse speculative decoding methods can all potentially serve as drafters in the pool, and we dynamically evaluate and select among them with provable no-regret guarantees.

## C.2 ADAPTIVE SPECULATIVE DECODING

Another line of work focuses on adapting speculative decoding during inference to incoming requests. OSD and OmniDraft (Liu et al., 2023a; Ramakrishnan et al., 2025) adapt the drafter on-the-fly to the target distribution via online knowledge distillation, improving token acceptance rate. Our method is training-free and operates at a different level: candidates in the drafter pool can themselves adopt such adaptive mechanisms, while our contribution lies in selecting among them in the multi-drafter setting. SpecDec++ (Huang et al., 2025a) instead adapts the speculation length, stopping drafting once the predicted rejection probability exceeds a threshold, which differs from our goal of multi-drafter selection. SpecServe (Huang et al., 2025b) takes a system-level perspective, adapting speculative decoding configurations (e.g. resource allocation) at runtime to meet latency and throughput SLOs, rather than focusing on acceptance probability. Among these works, MetaSD (Kim et al., 2024) first poses the problem and BanditSpec (Hou et al., 2025) represents the state-of-the-art for adaptive multi-drafter speculative decoding. In contrast, we advocate a different paradigm that cheaply exploits global information, achieving higher token acceptance rates and lower per-token latency.

## C.3 ONLINE LEARNING ALGORITHMS

Hedge and multi-armed bandits capture the full-information and partial-information settings, respectively, and have inspired numerous variants. The Hedge algorithm (Cesa-Bianchi & Lugosi, 2006) provides a classic framework for expert weighting; NormalHedge (Chaudhuri et al., 2009) removes the need for tuning learning rates; and AdaNormalHedge (Luo & Schapire, 2015) further improves adaptivity. In parallel, the stochastic $K$-armed bandit problem was introduced by Robbins (Robbins, 1952), leading to a wide family of exploration–exploitation algorithms. Canonical examples include UCB (Auer et al., 2002a) and KL-UCB (Garivier & Cappé, 2011), which provide confidence-based exploration guarantees, and EXP3 (Auer et al., 2002b), which provides regret guarantees in the adversarial setting. As compared in Section D.6, adopting Hedge and its variants consistently achieves strong performance in our experiments, surpassing bandit-style algorithms. This highlights the effectiveness of leveraging global loss information across experts.

# D ADDITIONAL EXPERIMENTAL DETAILS AND RESULTS

## D.1 EAGLE'S DEFAULT GENERATION CONFIGURATION

In our experiments, we directly adopt EAGLE's generation configuration as our framework focuses on the drafter selection problem. By default, EAGLE uses an exploration depth of 7, resulting in a speculative decoding length of 9 tokens per chunk, with the top-k value equals to 10 during its expanding and reranking phase.

## D.2 BERT CLASSIFIER TRAINING DETAIL

Table 5 records the training hyperparameters of the offline BERT classifier for statically routing the requests to the corresponding drafter. The datasets used for training the classifier is the aggregation of data across all 7 domains.

| Hyperparameter | Value |
|---|---|
| learning_rate | 2e-5 |
| per_device_train_batch_size | 128 |
| per_device_eval_batch_size | 128 |
| num_train_epochs | 3 |
| lr_scheduler_type | linear |
| warmup_ratio | 0.1 |
| optimizer | adamw |

Table 5: Training hyperparameters for BERT classifier.

## D.3 STATISTICS FOR THE CURATED QWEN DRAFTERS

Table 6 and 7 shows statistics of the 7 curated drafters with Qwen-3-8B and Qwen-3-32B as the target (**bold** indicates the best). Each drafter performs well in-domain (diagonal) but degrades when applied out-of-domain for Qwen-3-8B model, and on average is weaker than the vanilla EAGLE model. Similar patterns are shown in Table 1. Together, these drafters provide a realistic evaluation pool for HedgeSpec, which orchestrates them to jointly accelerate serving.

| Datasets | Python | | Math | | Biology | | Chemistry | | MedQA | | CNN_DM | | SQL | | | |
|---|---|---|---|---|---|---|---|---|---|---|---|---|---|---|---|---|
| Drafter domains | MAT | Token/s | MAT | Token/s | MAT | Token/s | MAT | Token/s | MAT | Token/s | MAT | Token/s | MAT | Token/s | Avg MAT | Avg Token/s |
| Vanilla Eagle | 4.96 | 55.84 | 4.52 | 50.88 | 4.06 | 46.22 | 3.98 | 44.85 | 3.84 | 44.13 | 4.07 | 46.20 | 4.20 | 44.60 | **4.23** | **47.53** |
| Python | **6.43** | **73.73** | 4.50 | 51.82 | 2.46 | 26.85 | 2.97 | 34.41 | 2.36 | 26.81 | 2.45 | 28.26 | 3.85 | 44.42 | 3.57 | 40.90 |
| Math | 4.23 | 50.50 | **7.39** | **86.33** | 2.66 | 31.67 | 3.46 | 39.38 | 2.67 | 29.96 | 2.87 | 32.93 | 3.68 | 40.20 | 3.85 | 44.42 |
| Biology | 3.20 | 37.25 | 3.61 | 43.01 | **5.70** | **67.39** | 3.74 | 43.90 | 3.80 | 45.11 | 2.70 | 31.50 | 2.80 | 33.99 | 3.65 | 43.16 |
| Chemistry | 3.79 | 43.36 | 5.40 | 61.07 | 3.79 | 43.10 | **6.69** | **76.68** | 3.54 | 40.03 | 2.74 | 29.14 | 3.36 | 38.38 | 4.19 | 47.39 |
| MedicalQA | 3.66 | 42.85 | 4.05 | 46.23 | 4.07 | 46.86 | 4.01 | 46.66 | **6.17** | **73.51** | 2.94 | 33.38 | 3.42 | 39.65 | 4.05 | 47.02 |
| CNN_DM | 2.56 | 29.80 | 2.56 | 30.40 | 2.45 | 28.49 | 2.45 | 28.61 | 2.55 | 30.21 | **5.15** | **60.70** | 2.52 | 29.47 | 2.89 | 33.95 |
| SQL | 3.71 | 43.63 | 3.37 | 36.30 | 2.36 | 27.10 | 2.68 | 31.74 | 2.37 | 28.26 | 2.60 | 30.27 | **7.60** | **86.17** | 3.53 | 40.49 |

Table 6: Statistics of the 7 curated drafters with Qwen-3-8B as the target. **Bold** indicates the best. Each drafter shows strong in-domain performance (diagonally strong) but suffers noticeable inefficiency when applied outside, and on average performs worse than the vanilla Eagle model. Similar trends are observed in Table 1. These drafters form a realistic evaluation pool for evaluating HedgeSpec, which orchestrates them to jointly accelerate the serving process.

| Datasets | Python | | Math | | Biology | | Chemistry | | MedQA | | CNN_DM | | SQL | | | |
|---|---|---|---|---|---|---|---|---|---|---|---|---|---|---|---|---|
| Drafter domains | MAT | Token/s | MAT | Token/s | MAT | Token/s | MAT | Token/s | MAT | Token/s | MAT | Token/s | MAT | Token/s | Avg MAT | Avg Token/s |
| Vanilla Eagle | 3.02 | 21.98 | 3.36 | 24.16 | 2.62 | 19.30 | 3.01 | 21.53 | 2.57 | 18.33 | 2.59 | 18.67 | 3.00 | 21.34 | 2.88 | 20.76 |
| Python | **6.00** | **43.67** | 4.66 | 33.63 | 2.52 | 18.20 | 3.16 | 23.30 | 2.42 | 17.78 | 2.58 | 18.94 | 3.86 | 28.22 | 3.60 | 26.25 |
| Math | 4.04 | 29.75 | **7.12** | **48.67** | 2.75 | 20.10 | 3.64 | 26.54 | 2.74 | 20.33 | 3.00 | 21.98 | 3.79 | 27.61 | 3.87 | 27.85 |
| Biology | 3.18 | 22.04 | 3.88 | 28.73 | **5.98** | **43.10** | 4.04 | 29.62 | 3.90 | 28.87 | 2.81 | 20.11 | 2.89 | 22.20 | 3.81 | 27.81 |
| Chemistry | 3.29 | 23.71 | 5.32 | 37.27 | 3.91 | 28.72 | **6.61** | **47.26** | 3.56 | 25.66 | 2.66 | 18.63 | 2.96 | 21.06 | 4.04 | 28.90 |
| MedicalQA | 3.56 | 25.97 | 4.18 | 32.85 | 4.24 | 30.36 | 4.21 | 31.26 | **5.95** | **43.34** | 3.10 | 22.42 | 3.52 | 26.83 | **4.11** | **30.43** |
| CNN_DM | 2.54 | 18.78 | 2.56 | 18.13 | 2.54 | 18.39 | 2.56 | 18.66 | 2.61 | 18.50 | **5.22** | **36.18** | 2.57 | 19.16 | 2.94 | 21.11 |
| SQL | 3.58 | 20.69 | 3.54 | 25.95 | 2.42 | 17.73 | 2.80 | 20.26 | 2.43 | 17.84 | 2.75 | 20.09 | **7.25** | **52.73** | 3.54 | 25.04 |

Table 7: Statistics of the 7 curated drafters with Qwen-3-32B as the target. **Bold** indicates the best.

## D.4 EVALUATING HEDGESPEC UNDER THE NON-GREEDY INFERENCE SETTING

Table 8 shows the MAT (Mean Accepted Tokens) and Token/s (token generation rate) across datasets for each method under non-greedy target model inference. The temperature is set to be 0.5 in

| Datasets | Python | | Math | | Biology | | Chemistry | | MedQA | | CNN_DM | | SQL | | | |
|---|---|---|---|---|---|---|---|---|---|---|---|---|---|---|---|---|
| Methods | MAT | Token/s | MAT | Token/s | MAT | Token/s | MAT | Token/s | MAT | Token/s | MAT | Token/s | MAT | Token/s | Avg MAT | Avg Token/s |
| *LLaMA-3.1-8B-IT* | 1.00 | 17.77 | 1.00 | 17.97 | 1.00 | 18.03 | 1.00 | 17.35 | 1.00 | 18.81 | 1.00 | 18.41 | 1.00 | 18.25 | 1.00 | 18.08 |
| Vanilla Eagle | 6.48 | 83.75 | 5.78 | 74.52 | 5.74 | 74.41 | 5.04 | 62.61 | 4.65 | 58.82 | 5.06 | 62.55 | 5.93 | 77.99 | 5.53 | 70.66 |
| UCBSpec | 5.17 | 64.67 | 5.50 | 72.61 | 4.90 | 64.53 | 4.89 | 61.11 | 4.11 | 53.79 | 3.72 | 45.40 | 5.39 | 66.85 | 4.81 | 61.28 |
| EXP3Spec | 4.93 | 62.63 | 5.33 | 69.60 | 4.67 | 60.28 | 4.76 | 62.62 | 4.11 | 52.23 | 3.82 | 46.25 | 5.07 | 64.06 | 4.67 | 59.67 |
| **HedgeSpec** | **7.16** | **87.38** | **7.32** | **88.58** | **6.67** | **82.22** | **6.63** | **80.05** | **6.01** | **72.65** | **5.58** | **65.11** | **7.89** | **95.81** | **6.75** | **81.69** |
| *Qwen-3-8B* | 1.00 | 15.33 | 1.00 | 15.08 | 1.00 | 14.95 | 1.00 | 14.43 | 1.00 | 14.70 | 1.00 | 14.70 | 1.00 | 15.14 | 1.00 | 14.90 |
| Vanilla Eagle | 4.88 | 53.03 | 4.58 | 50.59 | 4.07 | 47.13 | 3.90 | 44.10 | 3.76 | 42.89 | 4.05 | 45.47 | 4.18 | 48.06 | 4.20 | 47.32 |
| UCBSpec | 4.45 | 50.84 | 5.25 | 56.77 | 4.07 | 45.80 | 4.54 | 47.68 | 4.12 | 41.97 | 3.52 | 39.43 | 5.19 | 56.55 | 4.45 | 48.43 |
| EXP3Spec | 4.34 | 48.00 | 5.20 | 55.19 | 3.88 | 43.07 | 4.32 | 44.75 | 3.95 | 40.96 | 3.33 | 36.94 | 4.95 | 54.09 | 4.28 | 46.14 |
| **HedgeSpec** | **5.82** | **62.63** | **7.09** | **76.90** | **5.56** | **60.80** | **6.40** | **68.77** | **5.84** | **63.67** | **4.96** | **53.50** | **7.12** | **76.75** | **6.11** | **66.15** |
| *Qwen-3-32B* | 1.00 | 8.52 | 1.00 | 8.57 | 1.00 | 8.35 | 1.00 | 8.56 | 1.00 | 8.38 | 1.00 | 8.51 | 1.00 | 8.82 | 1.00 | 8.53 |
| Vanilla Eagle | 3.03 | 22.91 | 3.36 | 25.18 | 2.52 | 19.25 | 2.99 | 22.40 | 2.55 | 18.68 | 2.58 | 19.31 | 3.06 | 23.04 | 2.87 | 21.54 |
| UCBSpec | 4.12 | 30.36 | 5.22 | 38.04 | 4.08 | 30.14 | 4.60 | 31.76 | 4.06 | 29.09 | 3.45 | 24.97 | 4.78 | 34.90 | 4.33 | 31.32 |
| EXP3Spec | 4.01 | 28.27 | 4.98 | 36.22 | 3.96 | 28.04 | 4.35 | 30.13 | 3.92 | 27.60 | 3.37 | 24.57 | 4.54 | 33.54 | 4.16 | 29.77 |
| **HedgeSpec** | **5.31** | **36.86** | **6.71** | **47.37** | **5.44** | **38.54** | **6.23** | **42.77** | **5.46** | **39.29** | **4.78** | **33.48** | **6.58** | **46.24** | **5.79** | **40.65** |

Table 8: MAT (Mean Accepted Tokens) and Token/s (token generation rate) across datasets for each method under non-greedy target model inference. **Bold** indicates the best. The trend is consistent with results in greedy setting showing in Table 3.

| Datasets | Python | Math | Biology | Chemistry | MedQA | CNN_DM | SQL |
|---|---|---|---|---|---|---|---|
| Joint trained model | 7.03 | **7.76** | 6.63 | **7.16** | 6.22 | 5.35 | 7.39 |
| Eagle | 6.48 | 5.88 | 5.95 | 5.28 | 4.96 | 5.31 | 5.99 |
| HedgeSpec | **7.69** | 7.69 | **7.18** | 7.10 | **6.47** | **5.88** | **8.06** |

Table 9: Comparison of MAT across domains for a jointly trained model, the vanilla EAGLE, and HedgeSpec. Bold indicates the best performance in each domain.

this experiment. The trend is consistent with Table 3. HedgeSpec consistently outperforms other baselines across different model sizes and domains, demonstrating improved acceptance efficiency and throughput through adaptive orchestration of expert drafters.

## D.5 DISCUSSION ON HEDGESPEC VS. JOINTLY TRAIEND DRAFTER

In this section, we study the effect of aggregating data from all domains and jointly finetuning a EAGLE model. This effectively turns the original EAGLE into another "generic" drafter. Results are presented in Table 9, where HedgeSpec outperforms most domains in terms of MAT. In the meantime, several additional observations emerge:

First, joint training empirically improves EAGLE's performance across multiple domains, which is expected. Second, the jointly trained drafter performs comparably to expert drafters trained in single domain such as math, biology, chemistry, and MedQA, while we do observe performance drop in domains like python, CNN_DM, SQL in this experiment. The reasons require further investigation, but it is also reasonable to expect this since datasets from different domains may not be fully aligned, with some pushing learning in the same direction while others pull in the opposite (Liu et al., 2024); meanwhile, performance is also constrained by scaling laws and the model's capacity to digest knowledge—particularly given that EAGLE is only a one-layer transformer. As more data is introduced, it is uncertain whether the performance in those domains will continue to boost or decline. Third, even when proper joint training can yield synergistic benefits, it is often infeasible because parties may not release their training data due to confidentiality concerns (Achiam et al., 2023). In such cases, only the trained drafters might be available, making joint training impossible. This underscores HedgeSpec's advantage: it requires no access to training data and can operate directly on a pool of expert drafters. Moreover, a generically trained model can itself serve as one candidate within this pool alongside specialized drafters. In short, HedgeSpec addresses the higher-level challenge of orchestrating expert drafters. With only a collection of such models, it can significantly improve serving efficiency.

**Initially:** Set $R_{i,0} = 0$, $p_{i,1} = 1/N$ for each $i$.

**For** $t = 1, 2, \ldots$

1. Each action $i$ incurs loss $\ell_{i,t}$.

2. Learner incurs loss $\ell_{A,t} = \sum_{i=1}^{N} p_{i,t} \ell_{i,t}$.

3. Update the cumulative regrets: $R_{i,t} = R_{i,t-1} + (\ell_{A,t} - \ell_{i,t})$ for each $i$.

4. Find $c_t > 0$ that satisfying

$$\frac{1}{N} \sum_{i=1}^{N} \exp\left(\frac{([R_{i,t}]_+)^2}{2c_t}\right) = e.$$

5. Update distribution for round $t + 1$:

$$p_{i,t+1} \propto \frac{[R_{i,t}]_+}{c_t} \exp\left(\frac{([R_{i,t}]_+)^2}{2c_t}\right) \quad \text{for each } i.$$

Figure 7: The workflow of NormalHedge algorithm.

| Datasets | Python | Math | Biology | Chemistry | MedQA | CNN_DM | SQL |
|---|---|---|---|---|---|---|---|
| HedgeSpec | **7.69** | **7.69** | **7.18** | **7.10** | **6.47** | **5.88** | 8.06 |
| HedgeSpec w/ Acc. rate loss | 7.62 | 7.47 | 7.02 | 6.94 | 6.44 | **5.88** | **8.08** |
| HedgeSpec w/ Standard Hedge | 6.90 | 6.67 | 6.53 | 6.19 | 5.38 | 4.68 | 7.05 |
| HedgeSpec w/ AdaNormalHedge | 7.36 | 7.30 | 7.05 | 6.66 | 5.85 | 5.40 | 7.71 |

Table 10: Llama Mean Accepted Tokens (MAT) across datasets using HedgeSpec variants. Bold indicates highest performance per column.

## D.6 VARIANTS OF HEDGING ALGORITHMIC CHOICE

In this section, we study how different hedging algorithmic choices affect the final outcome. The ablation considers two factors: (i) using token acceptance–rate loss instead of expected-length–based loss, and (ii) replacing the NormalHedge update rule with alternative algorithms, including Standard Hedge and AdaNormalHedge. HedgeSpec by default adopts the parameter-free NormalHedge with expected-length–based loss. The MAT results across datasets are reported in Table 10 and 11, showing consistent trends. We below show the workflow of NormalHedge in Figure 7.

From the table, we see that the acceptance–rate–based loss achieves comparable but slightly lower MAT, suggesting it can be a viable alternative for online learning in speculative decoding, though expected-length–based loss remains stronger overall. Switching to Standard Hedge leads to a larger drop in performance. We attribute this to Standard Hedge's more conservative updating: it tends to spread weight more broadly during the exploration phase, whereas NormalHedge shrinks weights more aggressively toward strong drafters. This conservatism could be mitigated with careful parameter tuning, although such tuning would vary across scenarios and adds practical complexity. Finally, AdaNormalHedge also shows reduced MAT. This is somewhat surprising, as AdaNormalHedge enjoys first-order regret bounds and is theoretically stronger. In our experiments, however, this advantage did not materialize.

## D.7 ADDITIONAL RESULTS ON CUMULATIVE REGRET AND MAT VS. DRAFTER POOL TRENDS

In this section, we present additional results on cumulative regret for the Qwen-3-8B model, as well as results on MAT with an increasing number of drafters. The trends in Figure 8 and Figure 9 are consistent with those in Figure 5, demonstrating that HedgeSpec converges more quickly to near-zero regret and scales effectively as the drafter pool grows.

| Datasets | Python | Math | Biology | Chemistry | MedQA | CNN_DM | SQL |
|---|---|---|---|---|---|---|---|
| HedgeSpec | **6.32** | **7.27** | **5.66** | **6.61** | **6.08** | **5.10** | **7.52** |
| HedgeSpec w/ Acc. rate loss | 6.23 | 7.00 | 5.58 | 6.58 | 6.05 | 5.00 | 7.30 |
| HedgeSpec w/ Standard Hedge | 5.85 | 6.64 | 5.13 | 5.88 | 5.47 | 4.46 | 7.01 |
| HedgeSpec w/ AdaNormalHedge | 6.26 | 7.12 | 5.57 | 6.39 | 5.94 | 4.99 | 7.40 |

Table 11: Qwen-3-8B: MAT across datasets using HedgeSpec variants. Bold indicates highest per column.

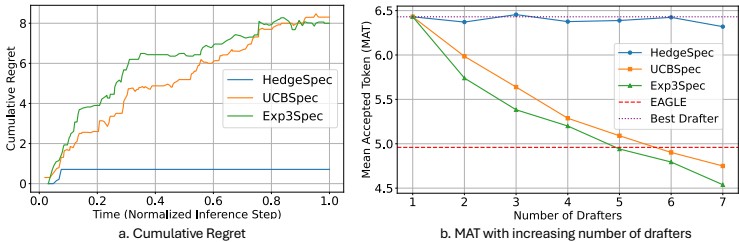

Figure 8: Cumulative regret and MAT vs. number of drafters. HedgeSpec quickly settles with near-zero regret, and can scale up with larger drafter pool. Llama results show similar trend in 5, highlighting our robustness.

## D.8   THE GSM8K, HUMANEVAL, BIRD, MULTINEWS AND SPECBENCH RESULTS

In this section, we present MAT and tokens-per-second results on the GSM8K, HumanEval, Bird, MultiNews datasets as shown in Table 12. These datasets are not included in the training data used to construct the drafters. We find that even without prior knowledge, HedgeSpec consistently outperforms all baselines across both domains. This demonstrates that HedgeSpec's orchestration of drafters can improve MAT and throughput without requiring prior knowledge, as long as some drafters excel in specific domains—highlighting HedgeSpec's effectiveness in drafter selection.

In Table 13, we provide additional results from running the SpecBench experiments. We note, however, that the experimental setup differs slightly from our main evaluation. SpecBench contains a substantial portion of tasks such as Natural Questions QA, translation, and RAG-style generation. Our drafter pool does not include experts specialized for these tasks—although the online learner can still select the "best available" drafter for each query.

However, this actually reflects the real scnario, that you could have a bunch of expert drafters, but you cannot cover all situation. In this sense, you usually have some additional "generalist" model other than the experts, like the EAGLE model, to handle the requests that outside the experts' expertise. With that being said, we added official EAGLE model in pool, and form up a 7 experts + 1 generalists pool for speculative decoding in this benchmark.

The results are shown below, we see that HedgeSpec still yeilds the best performance across different model. This experiments confirms HedgeSpec's effectiveness in selecting best drafters, regardless of OOD, for competing the best drafter in hindsight, leading to performance gain.

## D.9   DISCUSSION OF HEDGESPEC ON BROADER SPECULATIVE DECODING SCHEMES

In this section, we extend our discussion on the integration of HedgeSpec into broader speculative decoding schemes. We do believe HedgeSpec is integrable for different speculative decoding frameworks. At its core, HedgeSpec operates at an orthogonal layer: any speculative decoding algorithm could serve as a potential drafters in the pool. Theoretically, as long as we have a verified trace (Theorem 3), and can use it to get the token acceptance probability for EAL evalaution on other drafters, HedgeSpec can select among them with provable no-regret guarantees.

In our main experiments, we build upon EAGLE framework, which is widely regarded as the most broadly deployed framework (Marques et al., 2025; Tang et al., 2025). Across benchmarks, EAGLE

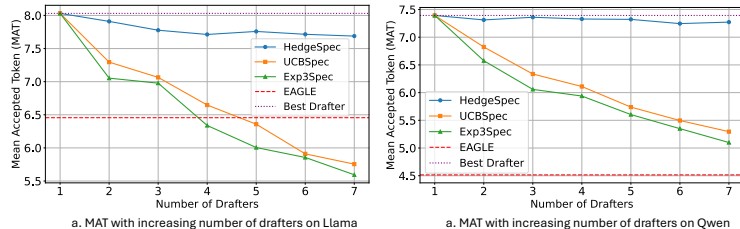

Figure 9: MAT vs. number of drafters on math workload on Llama and Qwen models. HedgeSpec scales effectively with larger drafter pools

| Datasets | GSM8K | | HumanEval | | Bird | | MultiNews | |
|---|---|---|---|---|---|---|---|---|
| **Method** | MAT | Token/s | MAT | Token/s | MAT | Token/s | MAT | Token/s |
| Llama-3.1-8B | 1.00 | 18.26 | 1.00 | 18.67 | 1.00 | 17.95 | 1.00 | 18.26 |
| EAGLE | 6.38 | 78.32 | 6.77 | 91.98 | 5.50 | 73.24 | 5.21 | **66.10** |
| EXP3Spec | 5.03 | 63.28 | 5.24 | 68.96 | 4.38 | 58.03 | 3.65 | 47.57 |
| UCBSpec | 5.05 | 66.11 | 5.46 | 72.05 | 4.67 | 62.60 | 3.58 | 46.04 |
| **HedgeSpec** | **6.77** | **84.32** | **7.54** | **97.21** | **6.67** | **83.87** | **5.53** | 65.20 |
| Qwen-3-8B | 1.00 | 14.82 | 1.00 | 14.71 | 1.00 | 14.91 | 1.00 | 14.89 |
| EAGLE | 5.05 | 60.59 | 4.84 | 58.81 | 4.08 | 47.91 | 4.00 | 46.81 |
| EXP3Spec | 4.90 | 53.57 | 4.15 | 48.62 | 3.73 | 43.67 | 3.21 | 36.26 |
| UCBSpec | 5.02 | 57.42 | 4.30 | 50.59 | 3.94 | 45.48 | 3.28 | 36.97 |
| **HedgeSpec** | **6.60** | **73.59** | **5.63** | **64.51** | **5.18** | **56.12** | **4.76** | **51.33** |
| Qwen-3-32B | 1.00 | 8.24 | 1.00 | 8.32 | 1.00 | 8.43 | 1.00 | 8.34 |
| EAGLE | 3.44 | 24.86 | 2.97 | 20.35 | 2.95 | 21.50 | 2.63 | 19.06 |
| UCBSpec | 5.00 | 37.50 | 3.97 | 29.66 | 3.85 | 27.56 | 3.48 | 25.09 |
| EXP3Spec | 4.86 | 36.38 | 3.88 | 28.62 | 3.78 | 27.36 | 3.34 | 23.46 |
| **HedgeSpec** | **6.57** | **42.37** | **5.17** | **33.82** | **4.92** | **34.72** | **4.84** | **33.25** |

Table 12: MAT (Mean Accepted Tokens) and Token/s (token generation rate) across GSM8K, HumanEval, Bird and MultiNews, which is not part of the training datasets for the drafters. **Bold** indicates the best. Additional results regarding SpecBench is shown in Table 13.Consistent with Table 3, HedgeSpec consistently outperforms all baselines across those two domains. Its shows that its orchestration of drafters improves MAT and throughput without prior knowledge, as long as there exists drafters excel in certain domain.

showS its SOTA performance over well-established speculative framework (i.e. Medusa, Rest, PLD). We believe that integrating HedgeSpec with EAGLE provides a strong and representative demonstration of our method's effectiveness. Below, we further discuss how HedgeSpec can be integrated into these other frameworks as well.

Regarding PLD (Prompt-Lookup-Decoding) (Saxena, 2023): PLD utilizes the input prompt to guess the generation. In some tasks with input grounded generation, there are high chances of overlap between input and output. As a result, PLD uses only a single surrogated drafter. Therefore, PLD is not the best candidates for the multi-drafter framework.

Regarding Medusa (Cai et al., 2024): Medusa is a well-known speculative decoding method. It utilizes multiple Medusa heads to predict the future tokens. These heads require training in order to align with the target task distribution. A potential integration with Medusa would follow a procedure similar to our EAGLE setup, that it obtains logits from the different set of expert Medusa heads and use them for full-information evaluation. This allows HedgeSpec to orchestrate the Medusa experts during speculative drafting.

Regarding REST (Retrieval-Based Speculative Decoding) (He et al., 2023): REST is another well-known speculative decoding framework. It treats different datastore as the surrogate drafters, gen-

| | Llama-3.1 | | | | Qwen-3-8B | | | | Qwen-3-32B | | | |
|---|---|---|---|---|---|---|---|---|---|---|---|---|
| | Vanilla | EXP3 | UCB | Hedge | Vanilla | EXP3 | UCB | Hedge | Vanilla | EXP3 | UCB | Hedge |
| **MAT** | 1.00 | 4.04 | 4.05 | **5.31** | 1.00 | 3.58 | 3.65 | **4.70** | 1.00 | 3.43 | 3.38 | **3.97** |
| **Token/s** | 18.24 | 52.19 | 52.68 | **60.80** | 14.32 | 41.02 | 41.23 | **50.70** | 8.12 | 24.86 | 24.24 | **27.73** |

Table 13: MAT and Token/s on SpecBench across different models (Llama-3.1, Qwen-3-8B, Qwen-3-32B) under various speculative decoding strategies. Best results are **bolded**.

| Model | HumanEval | | GSM8K | | MT-Bench | | Alpaca | |
|---|---|---|---|---|---|---|---|---|
| | MAT | Token/s | MAT | Token/s | MAT | Token/s | MAT | Token/s |
| Vicuna-7B-v1.5 | 1.00 | 33.59 | 1.00 | 33.49 | 1.00 | 33.48 | 1.00 | 33.31 |
| UCBSpec | 1.45 | 44.60 | 1.38 | 42.33 | 1.10 | 37.86 | 1.38 | 43.85 |
| Exp3Spec | 1.43 | 43.55 | 1.36 | 41.99 | 1.11 | 38.40 | 1.44 | 46.63 |
| **HedgeSpec** | **1.88** | **55.27** | **1.62** | **46.58** | **1.28** | **41.42** | **1.69** | **52.87** |

Table 14: MAT and Token/s across HumanEval, GSM8K, MT-Bench, and Alpaca for HedgeSpec comparing with other Bandit-based method on REST framework. **Bold** indicates the best.

erating future tokens by retrieving and composing relevant content from a reservoir of existing knowledge. To fit into a practical multi-drafter selection framework, we can construct multiple such expert reservoirs, each specializing in a particular domain for token drafting. To integrate with HedgeSpec, we can calculate the token acceptance probability (EAL) from each drafter with the verified trace, and use it to select the best drafter fitting the requests.

We integrate HedgeSpec into the REST framework. Following the original REST setup, we use Vicuna-7B-v1.5 as the target model. The expert drafters in this experiment include Python, Math, CNN-DM, MedQA, Alpaca, SQL, and Biology. Our goal remains the same: given a pool of diverse drafters, determine how to select the best drafter for each incoming task. We benchmark performance on HumanEval, GSM8K, MT-Bench, and Alpaca, with results shown in Table 14. Similar to our findings under the EAGLE framework, HedgeSpec again achieves the best performance. This demonstrates the advantage of leveraging full-information feedback in drafter selection, allowing HedgeSpec to adapt to the best drafter in hindsight and ultimately improving overall efficiency.

# E  ETHICS STATEMENT

This paper presents work with the goal of advancing machine learning. There are many potential societal consequences of our work, none of which we feel must be specifically highlighted here.

# F  REPRODUCIBILITY STATEMENT

The models and datasets used in this paper are fully open-sourced, with specifications provided in Sections 4.1 and 4.2. Additional configuration details and implementation specifics are referenced in the main text and included in Appendix B.5, D.1, and D.2. For theoretical contributions, complete proofs of the theorems are provided in Appendix A. Together, these materials ensure the reproducibility of our results.

