# OpenReview forum: "Not-a-Bandit: Provably No-Regret Drafter Selection in Speculative Decoding for LLMs"
_ICLR.cc/2026/Conference — ICLR 2026 Poster_

### Official Review · Reviewer_PCBy · 2025-10-26

**Soundness:** 3
**Presentation:** 2
**Contribution:** 3
**Rating:** 4
**Confidence:** 4

**Summary:**

This paper considers draft model selection in speculative decoding. It designs an online learning algorithm, HedgeSpec, which chooses the draft model in an online manner with full-information feedback, i.e., all the draft models can be evaluated without consuming much additional time. Theoretical guarantees for the token acceptance probability and expected acceptance length are provided, compared to the best fixed draft model in hindsight. Experiments are carried out with Llama-3.1-8B and Qwen-3-8B as the target models, and Eagle 3 series as the draft models.

**Strengths:**

1. The idea of using the verified trajectory to evaluate all the draft models is novel. It circumvents the bandit feedback setup in the literature and provides an alternative full-information with delayed feedback approach.
2. It provides some theoretical guarantees on the performance of HedgeSpec for both acceptance probability and expected acceptance length objectives.
3. Empirical experiments with Eagle 3 are conducted to demonstrate the effectiveness of the proposed approach.

**Weaknesses:**

1. **More clarity:** Some definitions of the notations are unclear. For example, do the two bounds in Theorem 4 hold at the population level or the prompt level? Are the probabilities conditional on $x_{<t}$ or on the entire input prompt? Is $T_{\text{token}}$ fixed, or is it a stopping time? If it is fixed, what happens when the generated sequence terminates before $T_{\text{token}}$? If it is a stopping time, how does the expected result look?

2. **Lack of interpretation:** More interpretation of the regret results would be helpful. In particular, for the result on accepted length (Line 299), while _AcceptLength_ is positively correlated with the quality of the draft model, its practical meaning is not well elaborated. For instance, the algorithm initiates speculative decoding only on a subsequence of the token sequence, and this subsequence is random.

3. **Limited draft methods:** The experiments are conducted exclusively with the EAGLE-3 series, providing limited evidence for the claim in Line 66 that “HedgeSpec could work with any given set of drafters on any speculative decoding method.”

4. **Mismatch between theory and experiments:** The theoretical results are derived under an adversarial setting, whereas the empirical studies are conducted under an i.i.d. assumption (Line 318). While parameter tuning in experiments is understandable, the shift from adversarial to i.i.d. settings may be too large. Moreover, the motivation for adopting the i.i.d. assumption is unclear, since tokens in autoregressive generation depend on all previous tokens.

**Minors (mostly about writing)**:
- I suggest that the authors carefully review the notations throughout the manuscript to ensure consistency and correctness, which would significantly improve readability. A few examples I noticed are listed below:

    1. Inconsistent notation: Accept_Length, $T_{Token}$.

    2. Line 199: an extra “=” in $q_i$.

    3. Line 209: the conditional probability expression appears problematic.

    4. The use of $\gamma_{i,j}$.

    5. Line 760: $h$ should be $t$.


- For self-consistency, it would be better to include the algorithms used in the appendix—for example, Algorithms 1 and 2 from Joulani et al.—as the _BASE_ algorithm in Theorem 4 does not appear within this manuscript.

- Since the paper claims that drafter selection can be performed in a full-information setting, it would be useful to explicitly discuss the corresponding losses and gains from both theoretical and empirical perspectives. There is likely a trade-off between faster learning and computational cost, as evaluating more draft models inevitably increases time and resource consumption.

**Questions:**

1. The proof of Theorem 4 invokes Theorem 1 from Joulani et al. (2013), which requires the delays to be independent of the forecaster’s prediction. However, this assumption seems violated in the setup of this paper, as the delays depend on the generated tokens of the chosen drafter model. Could the authors clarify this point?
2. Could the authors elaborate on Line 52: “Hedge or NormHedge to identify the best drafter _exponentially_ faster in N than bandit-based approaches”? It would be helpful to specify in what sense the improvement is exponential.
3. According to Figure 2, does this imply that the statistics at one position are reused multiple times? Can the authors further clarify on how these statistics are shared or updated across positions?
4. As claimed in Appendix B.5, evaluating more draft models slightly increases latency in the inference system. How does this latency scale with the number of draft models N? If the increase becomes non-negligible for large N, would a semi-bandit feedback mode—evaluating only a subset of the draft models per step (between bandit and full-information mode)—be a more practical strategy?
5. If one relaxes the acceptance criterion on Line 96, does the proposed method still work? How to adapt the method to this scenario?

Please also refer to the weaknesses above.

I favor the delicate way to evaluate all draft models and would appreciate it if the authors can resolve my concerns.

---

> ### Author Response · Authors · 2025-11-24
> **Response to reviewer PCBy**
>
> We would like to thank the reviewer very much for favoring our evaluation mechanisms and acknowledging our advantages! We would like to respond and address the reviewer's question regarding **clarification on notation and acceptance length, HedgeSpec on different spec decoding framework, clarification on theory/exp mismatch, evaluation overhead discussion** and more.
>
> We also would like to high light some additional results regarding **larger model results(Qwen-3-32B), more benchmarking results on OOD datasets** if the reviewer might be interested to see!
>
> ## W1. `Clarification of the notation`:
>
> We thank the reviewer for this clarification question! Our result applies for each query (user prompt).  The stated results in Theorem 4 are for the full expectation over the distribution of the entire roll-out from that prompt using the target model.
>
> Regarding the dependence on T, since the algorithm we apply (NormalHedge) is an "Any Time" algorithm, its regret bound applies to every fixed T = 1, 2, 3, ....  simultaneously.
>
>
> ## W2. `Lack of interpretation & Clarification regarding Acceptance Length`:
>
> We apologize if we missed anything from the question. But the Expected AcceptLength is not just simply correlated with the quality of the draft model, it is arguably the right measure of a draft model's performance in speculative decoding. Speculative decoding accelerates inference because the correct guess (accepted tokens in each chunk) makes a single expensive target pass produce produce multiple tokens (a normal decoding step produces one token), thus reducing per-token latency. A method that gives an expected acceptance length of say 5.5 will cover on average 5.5 tokens using the time for generating 1 token in the sequential model (modulo additional system-level overhead).
>
> The reviewer is absolutely right taht the subsequence is random and our acceptance length is defined as E[accept lengh  |  subsequence]  but our guarantee is stated for E[ E[accept lengh  |  subsequence] ] when the expectation in the outside is over the randomness of the subsequences induced by the target model.

---

> > ### Author Response · Authors · 2025-11-24
> > **thanks**
> >
> > ## W3. `HedgeSpec on different spec decoding framework`: We believe HedgeSpec can work with other speculative decoding methods. We provide additional experimental evidence on REST framework, and discuss regarding other speculative decoding framework.
> >
> > We thank the reviewer for raising the discussion of HedgeSpec integration! We do believe HedgeSpec is integrable for different speculative decoding frameworks. At its core, HedgeSpec operates at an orthogonal layer: any speculative decoding algorithm could serve as a potential drafters in the pool. Theoretically, as long as we have a verified trace (Theorem3), and can use it to get the token acceptance probability for EAL evalaution on other drafters, HedgeSpec can select among them with provable no-regret guarantees.
> >
> > In our submission, we build upon EAGLE framework, which is widely regarded as the most broadly deployed framework[1, 2, 3, 4]. Across benchmarks, EAGLE showS its SOTA performance over well-established speculative framework (i.e. Medusa, Rest, PLD). We believe that integrating HedgeSpec with EAGLE provides a strong and representative demonstration of our method’s effectiveness. Below, we further discuss how HedgeSpec can be integrated into these other frameworks as well.
> >
> > Regarding PLD (Prompt-Lookup-Decoding)[5]: PLD utilizes the input prompt to guess the generation. In some tasks with input grounded generation, there are high chances of overlap between input and output. As a result, PLD uses only a single surrogated drafter. Therefore, PLD is not the best candidates for the multi-drafter framework.
> >
> > Regarding Medusa[6]: Medusa is a well-known speculative decoding method. It utilizes multiple Medusa heads to predict the future tokens. These heads require training in order to align with the target task distribution. A potential integration with Medusa would follow a procedure similar to our EAGLE setup, that it obtains logits from the different set of expert Medusa heads and use them for full-information evaluation. This allows HedgeSpec to orchestrate the Medusa experts during speculative drafting.
> >
> >
> > Regarding REST (Retrieval-Based Speculative Decoding)[7]: REST is another well-known speculative decoding framework. It treats different datastore as the surrogate drafters, generating future tokens by retrieving and composing relevant content from a reservoir of existing knowledge. To fit into a practical multi-drafter selection framework, we can construct multiple such expert reservoirs, each specializing in a particular domain for token drafting. To integrate with HedgeSpec, we can calculate the token acceptance probability (EAL) from each drafter with the verified trace, and use it to select the best drafter fitting the requests.
> >
> >
> > We integrate HedgeSpec into the REST framework. Following the original REST setup, we use Vicuna-7B-v1.5 as the target model. The expert drafters in this experiment include Python, Math, CNN-DM, MedQA, Alpaca, SQL, and Biology. Our goal remains the same: given a pool of diverse drafters, determine how to select the best drafter for each incoming task. We benchmark performance on HumanEval, GSM8K, MT-Bench, and Alpaca, with results shown in the table below. Similar to our findings under the EAGLE framework, HedgeSpec again achieves the best performance. This demonstrates the advantage of leveraging full-information feedback in drafter selection, allowing HedgeSpec to adapt to the best drafter in hindsight and ultimately improving overall efficiency.
> >
> > | Model            | HumanEval MAT | HumanEval Token/s | GSM8K MAT | GSM8K Token/s | MT-Bench MAT | MT-Bench Token/s | Alpaca MAT | Alpaca Token/s |
> > |------------------|----------------|--------------------|-----------|----------------|----------------|--------------------|-------------|------------------|
> > | Vicuna-7B-v1.5   | 1.00           | 33.59              | 1.00      | 33.49          | 1.00           | 33.48              | 1.00        | 33.31            |
> > | UCBSpec          | 1.45           | 44.60              | 1.38      | 42.33          | 1.10           | 37.86              | 1.38        | 43.85            |
> > | Exp3Spec         | 1.43           | 43.55              | 1.36      | 41.99          | 1.11           | 38.40              | 1.44        | 46.63            |
> > | HedgeSpec        | **1.88**           |**55.27**              | **1.62**      | **46.58**         | **1.28**          | **41.42**           | **1.69**       | **52.87**         |
> >
> >
> > Table4, HedgeSpec performance under REST framework comparing with the Bandit based method. HedgeSpec achieves the best across datasets consistently, indicating its effectiveness.

---

> > > ### Author Response · Authors · 2025-11-24
> > > **thanks**
> > >
> > > [1] [Efficient Speculative Decoding for Llama at Scale: Challenges and Solutions](https://ai.meta.com/research/publications/efficient-speculative-decoding-for-llama-at-scale-challenges-and-solutions/)
> > >
> > > [2] [Standardized, production-ready speculative decoding](https://developers.redhat.com/articles/2025/11/19/speculators-standardized-production-ready-speculative-decoding)
> > >
> > > [3] [NVIDIA NeMo Framework](https://docs.nvidia.com/nemo-framework/user-guide/latest/model-optimization/speculative/speculative.html)
> > >
> > > [4] [NVIDIA TensorRT LLM](https://github.com/NVIDIA/TensorRT-LLM/tree/main/examples/eagle)
> > >
> > > [5] [Prompt-Lookup-Decoding](https://github.com/apoorvumang/prompt-lookup-decoding)
> > >
> > > [6] Medusa: Simple LLM Inference Acceleration Framework with Multiple Decoding Heads
> > >
> > > [7] REST: Retrieval-Based Speculative Decoding
> > >
> > > ## W4. `Mismatch between theory and experiments`:
> > >
> > > We thank the reviewer for the question! Our problem is somewhat special. The setting is a Markov process as you said. It is not iid due to what you described, but also it is a bit too conservative in practice to adopt the fully-adversarial setting. In some sense, we can argue it is closer to an iid setting if the markov process mixes quickly.
> > >
> > > Theoretically, the iid setting gives you a regret bound that depends only additively on the expected delay rather than multiplicatively as in the adversarial case. Empirically, we find that the iid setting is a good approximation and the regret is very small (see Figure 4).
> > >
> > > We'd like to point out that existing work adopted this iid assumption too. In the BanditSpec paper (ICML'25), they used UCB-Spec which assumes the iid assumption.
> > >
> > > Bridging the gap in the regret bound between iid and adversarial case has been a theme of research in online learning, there are theoretical results that aim at getting the best of both worlds, e.g., in
> > > "Follow the Leader If You Can, Hedge If You Must" "https://arxiv.org/abs/1301.0534"
> > >
> > > It is not uncommon for people to report that the iid setting is a better approximation in practice and results in significantly faster learning.
> > >
> > > ## W5.`minor writing comments`: We will fix them!
> > >
> > > Thanks the reviewer for the careful read of our paper, we shall fix those comments, and we put the base algorithm (NormalHedge) into our paper [here](https://openreview.net/pdf?id=JMmljf895g#page=2). To make the paper more readable, we add an overall figure to our paper [here](https://openreview.net/pdf?id=JMmljf895g#page=23).
> > >
> > > ## Q1. `Clarification on "The assumption seems violated..."`:
> > >
> > > That is a very good question! Our main Theorem 4 is about the token-level game. In the token-level game, the time index is the token index (detached from the actually chunks that speculative decoding rolls out).  The delay is fixed at the maximum length of speculative decoding, thus independent to the actions.
> > >
> > > In the chunk-level game, the delay does depend on the action taken.  We did not state any theoretical results for the chunk-level games. The only theorem we stated about the chunk-level game is Theorem 6 in the appendix, in which we used the naive censored feedback, and there is no delay.
> > >
> > > We hope the above convinces the reviewer that there is no issue with soundness of our theoretical results.
> > >
> > > Just in case the reviewer is curious: We are able to obtain a regret bound with "delayed" rather than "censored but not delay" feedback for the chunk-level game. This was not stated anywhere in the paper, mostly because we didn't think it's important.
> > >
> > > If we use the delayed feedback as in the RHS of Figure 2 instead (in the chunk-level game), then we can still get a bound that depends on the fixed upper bound of the delay without the need of an independence assumption.  This follows from the bottom of the left column in Page 5 (proof of Theorem 1) in the paper by Joulani et al,  apply the adversarial regret bound of the BASE algorithm, then apply Cauchy-Shwartz inequality with the fixed upper bound of M_n.  Observe that no-independence is needed. If the reviewer insists, we can add this as a theorem in the appendix.
> > >
> > > ## Q2. `Elaboration of "Hedge or NormHedge is exponentially faster..."`:
> > >
> > > We thank the reviewer's question. We elaborate why NormalHedge is exponentially faster than bandit base method. Bandit methods give a regret bound proportional to $$\sqrt{T N}$$ while Hedge gives a regret bound of $$\sqrt{T \log N}$$
> > > From $poly(N)$ to $polylog(N)$ is an exponential improvement.

---

> ### Author Response · Authors · 2025-11-24
> **thanks**
>
> ## Q3. `Clarifications on "statistics at one position are reused multiple times in Figure 2"`:
>
> Yes, as shown in Figure2 in the paper, the statistics are reused multiple times in order to construct our Accept-Length estimators. For these reasons, the feedback that we get are correlated in complicated manners. This is one of the reasons our Theorem 4 is interesting, as it deals with these complex dependence, and show that we can still take expectation on both sides and get a result that is highly interpretable.
>
> (Hint: it is due to the martingale-type structure).
>
> ## Q4. `Discussion on the case with larger drafter pool`: Semi-bandit is a very interesting idea, and we have a potentially even more interesting thought to discuss! Please see it below.
>
> HedgeSpec does incur drafter evaluation for full information. Regarding the additional latency incurred during drafter evaluation, this evaluation step is theoretically parallelizable, since it depends only on the verified trajectory obtained after the target model verification stage. In practice, this requires some engineering for optimization. In our HedgeSpec experiments, we empirically show that the overhead is relative small: evaluating 7 drafters adds only ~2% latency for Qwen-3-8B and ~1.4% for Qwen-3-32B. As demonstrated in our main evaluation results, this minor cost is outweighed by the benefits of full-information feedback. HedgeSpec achieves higher token acceptance rates, which reduce the number of expensive target model forwards and ultimately lead to better overall efficiency.
>
> | Model        | Python | Math  |
> |--------------|--------|-------|
> | Qwen-3-8b    | 1.90%  | 2.00% |
> | Qwen-3-32b   | 1.20%  | 1.40% |
>
> Table5, drafter evaluation overhead (7 drafters in total) of Qwen target model under Python/Math workload.
>
> Regarding some practical solution for very large drafter pool. As the reviewer has suggested, the semi-bandit idea is very interesting.  You can get a variance of the feedback vector between O(1) and O(N). That is a potentially useful knob to tune in production systems.
>
> In the meantime, we do have a more ambitious idea in mind. Remember that we are already handling delays anyway, how about to make the "evaluation" completely detached, so as to compute asynchronously? To be more specific, you can imagine there is a background thread continuously evaluate the drafters, of course with delayed feedback. For the main thread, it conduct just the usual speculative decoding step, but keeps choosing the best drafter chosen from the background thread, and sent the background thread with the statistics (verified trajectory) in each step for computing the loss. With that, evaluation can be done asynchronously, and our delayed feedback analysis is directly applicable to this scenario, which ease the overhead of evaluation. This could be a solution regarding very large drafter pool, but still with full-information view.
>
> ## Q5. `"Does HedgeSpec still work in relaxed acceptance criterion?"`
>
> Line 96 is about the basic rule in speculative sampling and is important for speculative decoding to correctly sample from the target model (to preserve the distribution).  EAGLE uses greedy drafters to spawn a draft tree, thus trivially simplifying the logic here, while still retaining the correctness of the sample.
>
> Our proposed method can work with any speculative decoding method that correctly samples from the target model.
>
> ## More experiments regarding Qwen-3-32B model, HedgeSpec on more OOD data i.e. SpecBench:
>
> In the meantime, we do want to advertise some additional experimental results we have regarding
> 1. a full set of HedgeSpec experiment on Qwen-3-32B model [here](https://openreview.net/forum?id=JMmljf895g&noteId=mk9ekRBmwe),
> 2. more benchmarks on OOD data other than GSM8K and HumanEval on SpecBench, Bird and Multinews [here](https://openreview.net/forum?id=JMmljf895g&noteId=DIS6sEiNgh).
>
> We do proudly say that HedgeSpec's performance is effective and consistent under larger target model and different benchmark data. If you might be interested, please kindly refer to the corresponding discussion for more information!
>
> ## Updated manuscript:
>
> We have updated our manuscrits and we mark the updated text in **purple**. If you might be available, please take a look!
>
> ## Thanks again!
>
> We hope our response effectively addressed the reviewer's questions and concerns! If there is any further questions/comments, please feel free to let us know!

---

### Official Review · Reviewer_quC5 · 2025-10-31

**Soundness:** 3
**Presentation:** 3
**Contribution:** 2
**Rating:** 2
**Confidence:** 4

**Summary:**

This paper addresses adaptive drafter selection in speculative decoding for large language models. Instead of relying on a fixed small drafter, the authors propose HedgeSpec, an online learning framework that dynamically selects among multiple drafters during inference.

The key insight is that speculative decoding inherently provides full-information feedback—the target model’s verified trajectory allows computation of all drafters’ performance without additional target calls. This reframes the problem from a bandit to a full-information setting, enabling no-regret guarantees via standard Hedge algorithms.

**Strengths:**

1. The paper presents a clean and insightful reframing of drafter selection as a full-information online learning problem. This perspective simplifies speculative decoding analysis and allows provable no-regret guarantees without additional target calls。

2. The proposed HedgeSpec framework is efficiently realized and demonstrates consistent throughput and acceptance gains across multiple domains and model families (LLaMA-3.1-8B and Qwen-3-8B).

**Weaknesses:**

- Although the evaluation phase introduces minimal overhead per drafter, it assumes access to sufficient computational resources for parallel evaluation. On resource-constrained systems, this could become a bottleneck. Since the results are only reported on 8 A100 GPUs, it would be helpful to include experiments on different hardware configurations and a breakdown of evaluation time and memory usage as the number of drafters N increases.

- The theoretical analysis relies on the assumption that feedback for all drafters can be computed without bias from the verified trajectory. Any deviation from this assumption may affect empirical optimality, and the error may accumulate as decoding process proceeds. An ablation across different numerical precisions would strengthen this claim.

- The evaluation scope is somewhat limited. (1) The method is described as training-free (Line 1043), but the draft models are pre-trained on specific datasets; and (2) the experiments primarily reuse these datasets. Beyond GSM8K and HumanEval, a broader evaluation on benchmarks such as SpecBench would be valuable.

- The experiments use Llama-3.1-8B-Instruct and Qwen-3-8B as target models. Demonstrating results on larger models would make the findings more convincing.

- Including experiments with batched prompts would provide insight into the method’s effectiveness in realistic, production-like settings.

- It would be interesting to discuss or demonstrate how the proposed approach integrates with other speculative decoding methods such as PLD, Medusa, and SAM.

**Questions:**

See weakness

---

> ### Author Response · Authors · 2025-11-24
> **Response to reviewer quC5**
>
> We thank the reviewer for acknoledging the advantage of our paper! We would like to address the reviewer's comments regarding **HedgeSpec on different hardware configurations, numerical precisions, More OOD evaluation including SpecBench, larger target model and HedgeSpec on different Spec decoding mechanisms.**
>
> ## W1. `HedgeSpec under constrained resources`: please see the analysis and experiments of overhead below.
>
>
> We thank the reviewer for the consideration on evaluating on constrained GPU environments! We understand that although A100 servers have been widely deployed [12], it would be beneficial to analyzie and test our method in some restricted resources. With that, we would like to first show some analysis regarding resource needed, then show a set of experiments of running HedgeSpec on a single A100 GPU shown in below Table1.
>
> #### 1. `How much resources are needed for multi-drafter speculative decoding? And what does HedgeSpec add up the overhead?`
>
> Comparing with a general multi-drafter speculative decoding framework, HedgeSpec adds at most the KV cache of each drafter compare to bandit based method (32MB * number of drafters in 8k ctx length), which is very small relative to the target model. Such overhead is induced because the evaluation conducted for full-information view.
>
> We note that HedgeSpec will not cause large HBM/KV cache overhead, nor does it cause thrashing. In multi-drafter spec decoding, both the target model and all drafter models will already reside in memory regardless of whether HedgeSpec or a bandit method is used. During runtime, the drafter selected by a bandit method holds the KV cache for the full context seen so far and retains it thereafter. For HedgeSpec, it lloads the KV cache for all drafters at each evaluation step. This is why we say HedgeSpec “adds at most the KV cache of each drafter.” However, this overhead is minimal, as the drafter is much smaller than the target model (EAGLE-3 is only a single-layer transformer) by design to maximize speculative decoding acceleration. For clarity and conciseness, we present the HBM and KV-cache calculations with 8k context length along with the peak runtime memory utilization below.
>
> For more concise calculation for clarity, we here report the exact HBM and KV cache analysis under 8k context.
>
> For a Llama-3.1-8B-IT (32 layers) model, it occupies ~16GB for HBM in FP16. EAGLE-3 drafter, which has around 0.38B parameters (1 layer), requires 0.76G in HBM, which is significantly smaller than the target, aligns with the usual goal of keeping the drafter much smaller than the target.
> For the KV cache at 8k context length, the 8B Llama-3.1 model has in total 32 layer, 8 KV heads, each head with head_dim = 128, the hidden size is 4096. Thus, KV_per_layer = 2 * num_kv_heads * seq_len * head_dim, meaning that this induces ~1GB KV cache (2 * 8192 seq_len * 8 num_kv_heads * 128 head_dim * 32 layer*  2 bytes ) = 32MB per layer. With that, a 32 layer model has 1024MB = 1GB KV cache under 8k ctx length. Each drafter requires only 32 MB of KV cache at 8k context, which is far smaller than the target model’s KV cache. Consequently, HedgeSpec adds at most 32 MB per drafter of additional KV-cache overhead compared with bandit methods.
>
> With these components, the total runtime memory overhead for our setup with 7 drafters with Llama-3.1-8B-IT is approximately 16 GB (target model) + 7 * 0.76 GB (drafter models) + 1GB (target KV cache) + 7 * 32MB (drafter KV cache)  memory. During runtime, we profile the peak memory utilization in python workload and get a 29093MB result. This is not a large overhead for a multi-drafter speculative system which lies well within the 40/80 GB capacity of a single A100. As noted above, HedgeSpec’s additional KV cache footprint is 7 * 32MB, which is manageable because the drafter is much smaller than the target model for better acceleration.
>
> Regarding the additional latency incurred during drafter evaluation, this evaluation step is theoretically parallelizable, since it depends only on the verified trajectory obtained after the target model verification stage. In practice, this requires some engineering for optimization. In our HedgeSpec experiments, we empirically show that the overhead is relative small: evaluating 7 drafters adds only ~2% latency for Qwen-3-8B and ~1.4% for Qwen-3-32B. As demonstrated in our main evaluation results, this minor cost is outweighed by the benefits of full-information feedback. HedgeSpec achieves higher token acceptance rates, which reduce the number of expensive target model forwards and ultimately lead to better overall efficiency.
>
> | Model        | Python | Math  |
> |--------------|--------|-------|
> | Qwen-3-8b    | 1.90%  | 2.00% |
> | Qwen-3-32b   | 1.20%  | 1.40% |
>
> Table0, drafter evaluation latency overhead (7 drafters in total) of Qwen target model under Python/Math workload.

---

> > ### Author Response · Authors · 2025-11-24
> > **thanks**
> >
> > #### 2. `Running HedgeSpec in a constrained environment with a single A100 GPU.`
> >
> > Following the memory-overhead analysis above, we also conducted an additional experiment running HedgeSpec on a single A100 server. The results are reported in Table 1 below. As expected, the findings are fully consistent with those in the main paper: HedgeSpec remains effective even in this more resource-constrained environment, continuing to provide improved multi-drafter selection and end-to-end efficiency.
> >
> > | **Qwen-3-8B**             | Math MAT | Math Token/s | Python MAT | Python Token/s |
> > |------------------|----------|---------------|-------------|------------------|
> > | vanilla          | 1.00     | 20.06         | 1.00        | 20.43            |
> > | EAGLE            | 4.52     | 68.48         | 4.96        | 75.31            |
> > | UCBSpec          | 5.30     | 78.64         | 4.75        | 67.60            |
> > | Exp3Spec         | 5.10     | 76.23         | 4.54        | 66.90            |
> > | HedgeSpec        | **7.27**     | **98.76**         | **6.32**        | 90.68            |
> > |                  |          |               |             |                  |
> > | **Llama-3.1-8B-IT** |   Math MAT | Math Token/s | Python MAT | Python Token/s |
> > | vanilla          | 1.00     | 23.78         | 1.00        | 23.50            |
> > | EAGLE            | 5.88     | 94.65         | 6.48        | 101.23           |
> > | UCBSpec          | 5.75     | 91.50         | 5.44        | 91.45            |
> > | Exp3Spec         | 5.59     | 90.41         | 5.16        | 85.63            |
> > | HedgeSpec        | **7.69**     | **114.65**        | **7.69**        | **115.40**           |
> >
> > Table1, HedgeSpec's performance across models under Python/Math workload on **a single A100** GPU setting. **Bold** indicates the best. HedgeSpec consistently outperforms other baselines under constrained environment.
> >
> >
> > [12] More than Carbon: Cradle-to-Grave environmental impacts of GenAI training on the Nvidia A100 GPU
> >
> > ## W2- `Clarification on numerical stability`: numerical precision will not break/deviate the unbiased estimator assumption
> >
> > We believe the reviewer is asking about the numerical precision from floating-point representation of real numbers. These issues are handled usually at a very low-level by numerical analysis, which usually comes with stability and error composition guarantees. For readability purposes, theoretical guarantees in machine learning (and most of the applied mathematics) are usually stated for ideal real numbers. We are no exception.
> >
> > We do note that numerical precision will not break/deviate the unbiased estimator assumption. Theorem 3 works with real numbers computed using the logits of the target model and that of the draft models. The computation is lipschitz, and does not amplify small numerical errors,  so if those $\gamma$s are approximately correct, then our estimator remains approximately correct too e.g., with a numerical error on the order of 1e-20. The resulting multi-step composition of these numerical errors over many tokens will also follow from standard numerical analysis.
> >
> > To make it more concrete, if all estimators are perturbed adversarially by $\epsilon$,  then we just need to add $O(T \epsilon)$ to the regret bound.
> >
> > In deep learning models, people often use FP16 and even FP8 for inference.  These issues are independent to what we are doing since the numerical issues are the same whether or not we use HedgeSpec or not.  It does not seem reasonable for us to use ultra low-precision for representing the states of HedgeSpec, since it is only O(N) numbers to keep track of.
> >
> > On the otherhand, we conduct an ablation study regarding using FP16/FP32, and we found that both mode achieves similar token acceptance rate (so as MAT) for Llama-3.1-8B-IT and Qwen-3-8B models, for both EAGLE and HedgeSpec on Math and Python workload, which empirically demonstrates our arguments.
> >
> > | Model         | Precision | Python (HedgeSpec) | Python (EAGLE) | Math (HedgeSpec) | Math (EAGLE) |
> > |---------------|-----------|-------------------|--------------|-----------------|------------|
> > | Llama-3.1-8B-IT | fp16      | 7.69              | 6.48         | 7.69            | 5.88       |
> > | Llama-3.1-8B-IT | fp32      | 7.69              | 6.47         | 7.69            | 5.89       |
> > | Qwen-3-8B     | fp16      | 6.32              | 4.96         | 7.27            | 4.52       |
> > | Qwen-3-8B     | fp32      | 6.37              | 4.97         | 7.27            | 4.49       |
> >
> > Table2: MAT on Llama-3.1-8B-IT and Qwen-3-8B under Python and Math workload in different numerical precision
> >
> > We hope the above explanation and ablation study can address your concern regarding numeric stability!

---

> > > ### Author Response · Authors · 2025-11-24
> > > **thanks**
> > >
> > > ## W3-1: `Clarification on training-free`: Our arguments hold, HedgeSpec is indeed a training-free method for drafter selection.
> > >
> > > We thank the reviewer for the question. The purpose of training the drafters in our work is solely to construct a diverse drafter pool, enabling large-scale evaluation of our training-free drafter selection framework. Aligning with prior work on drafter selection (e.g., MetaSD [1] and BanditSpec [2]), HedgeSpec focuses on the problem of how to compete with the best drafter in a given pool for each request. The method itself is a pure online learning algorithm and does not require any prior knowledge or training. Therefore, the fact that we train drafters only to construct the evaluation pool does not affect our claim: HedgeSpec is entirely training-free method for drafter-selection.
> > >
> > > [1] A Unified Framework for Speculative Decoding with Multiple Drafters as a Bandit
> > > [2] BanditSpec: Adaptive Speculative Decoding via Bandit Algorithms
> > >
> > >
> > >
> > > ## W3-2: `Additional evaluation on OOD datasets`: We respectfully argue the current evaluation supports our claim, and we have some additional results on OOD datasets like SpecBench, Bird and MultiNews holding our claims.
> > >
> > > We thank the reviewer for the consideration in running additional benchmarks! We first respectfully argue that using held-out test sets in our evaluation does not conflict with our claims. Our goal is to study how to select and orchestrate the drafters within a given pool to best answer each request, and our experiments demonstrate that adaptive orchestration of expert drafters consistently improves MAT and end-to-end throughput across models and datasets.
> > >
> > > #### 1. `Benchmarking on Bird and MultiNews datasets`
> > >
> > > In the meantime, we are happy that the reviewer think GSM8K and HumanEval evaluation is useful! Here we run additional benchmark (Bird (SQL generation [3]) and MultiNews(Summerization) [4]) follow the exact setting of what we presented in our [main paper]()'s experiments on GSM8K/HumanEval. The trend remains consistent: on these OOD datasets, HedgeSpec still outperforms all baselines. This reinforces that HedgeSpec does not rely on any prior knowledge about the incoming query. Instead, it leverages full-information feedback to adapt on the fly for the best drafter. As a result, it delivers substantial gains over bandit methods, which ultimately translate into real end-to-end efficiency improvements.
> > >
> > > [3] [A Big Bench for Large-Scale Database Grounded Text-to-SQLs](https://bird-bench.github.io)
> > >
> > > [4] Multi-News: a Large-Scale Multi-Document Summarization Dataset and Abstractive Hierarchical Model
> > >
> > >
> > > |           | Bird MAT | Bird Token/s | MultiNews MAT | MultiNews Token/s |
> > > |----------------|-----------|---------------|----------------|--------------------|
> > > | Llama-3.1-8B-IT        | 1.00      | 17.95         | 1.00           | 18.26              |
> > > | EAGLE          | 5.50      | 73.24         | 5.21           | **66.10**          |
> > > | EXP3Spec       | 4.38      | 58.03         | 3.65           | 47.57              |
> > > | UCBSpec        | 4.67      | 62.60         | 3.58           | 46.04              |
> > > | **HedgeSpec**  | **6.67**  | **83.87**     | **5.53**       | 65.20              |
> > > |                |           |               |                |                    |
> > > | Qwen-3-8B        | 1.00      | 14.91         | 1.00           | 14.89              |
> > > | EAGLE          | 4.08      | 47.91         | 4.00           | 46.81              |
> > > | EXP3Spec       | 3.73      | 43.67         | 3.21           | 36.26              |
> > > | UCBSpec        | 3.94      | 45.48         | 3.28           | 36.97              |
> > > | **HedgeSpec**  | **5.18**  | **56.12**     | **4.76**       | **51.33**          |
> > > |                |           |               |                |                    |
> > > | Qwen-3-32B        | 1.00      | 8.43          | 1.00           | 8.34               |
> > > | EAGLE          | 2.95      | 21.50         | 2.63           | 19.06              |
> > > | EXP3Spec       | 3.78      | 27.36         | 3.34           | 23.46              |
> > > | UCBSpec        | 3.85      | 27.56         | 3.48           | 25.09              |
> > > | **HedgeSpec**  | **4.92**  | **34.72**     | **4.84**       | **33.25**          |
> > >
> > > Table3, HedgeSpec's performance across Bird and MultiNews datasets, which is OOD of the experts training data. The trend is consistent with the results for GSM8K/HumanEval, showcasing HedgeSpec's effectiveness for drafter selection.

---

> > > > ### Author Response · Authors · 2025-11-24
> > > > **thanks**
> > > >
> > > > #### 2. `Benchmarking on SpecBench`
> > > >
> > > > In the table below, we provide additional results from running the SpecBench experiments. We note, however, that the experimental setup differs slightly from our main evaluation. SpecBench contains a substantial portion of tasks such as Natural Questions QA, translation, and RAG-style generation. Our drafter pool does not include experts specialized for these tasks—although the online learner can still select the “best available” drafter for each query.
> > > >
> > > > However, this actually reflects the real scnario, that you could have a bunch of expert drafters, but you cannot cover all situation. In this sense, you usually have some additional "generalist" model other than the experts, like the EAGLE model, to handle the requests that outside the experts' expertise. With that being said, we added official EAGLE model in pool, and form up a 7 experts + 1 generalists pool for speculative decoding in this benchmark
> > > >
> > > >
> > > > The results are shown below, we see that HedgeSpec still yeilds the best performance across different model. This experiments confirms HedgeSpec's effectiveness in selecting best drafters, regardless of OOD, for competing the best drafter in hindsight, leading to performance gain.
> > > >
> > > > | Model          | Metric      | Vanilla | EXP3Spec | UCBSpec | HedgeSpec |
> > > > |----------------|-------------|---------|----------|---------|-----------|
> > > > | Llama-3.1-8B-IT   | **MAT**     | 1.00    | 4.04     | 4.05    | **5.31**  |
> > > > | Llama-3.1-8B-IT   | **Token/s** | 18.24   | 52.19    | 52.68   | **60.80** |
> > > > | Qwen-3-8B      | **MAT**     | 1.00    | 3.58     | 3.65    | **4.70**  |
> > > > | Qwen-3-8B      | **Token/s** | 14.32   | 41.02    | 41.23   | **50.70** |
> > > > | Qwen-3-32B     | **MAT**     | 1.00    | 3.43     | 3.38    | **3.97**  |
> > > > | Qwen-3-32B     | **Token/s** | 8.12    | 24.86    | 24.24   | **27.73** |
> > > >
> > > > Table4: HedgeSpec's performance SpecBench. The official EAGLE is consisted inside the drafter pool to serve as a "generalist" complementary to the experts. HedgeSpec still achieves the best efficiency across other drafter selection mechanism.
> > > >
> > > >
> > > > ## W4- `Experiments on Larger target model`: our results on Qwen-3-32B hold our claims!
> > > >
> > > > We thank the reviewer's comments on larger model experiments, and we would like to show our results on Qwen-3-32B model! We use the exact same setting from other model presented in the paper, and we additionally trained 7 drafters follow the same manner for Qwen-3-32B target model. From the numbers shown below, the results are fully consistent with those from the 8B experiments: HedgeSpec continues to benefit from panoramic full-information feedback, which reduces the number of target-model calls and yields better overall efficiency, which is robust to different model size. These findings align with our earlier arguments. In addition, we provide further OOD results on Qwen-3-32B (Bird, MultiNews, SpecBench), all of which exhibit the same trend, reaffirming HedgeSpec’s robustness across datasets and model scales.
> > > >
> > > > | Method     | Python MAT | Python Tok/s | Math MAT | Math Tok/s | Biology MAT | Biology Tok/s | Chemistry MAT | Chemistry Tok/s | MedQA MAT | MedQA Tok/s | CNN_DM MAT | CNN_DM Tok/s | SQL MAT | SQL Tok/s | Avg MAT | Avg Tok/s |
> > > > |------------|-------------|--------------|-----------|-------------|--------------|----------------|----------------|------------------|------------|--------------|--------------|----------------|----------|-------------|----------|-------------|
> > > > | Qwen-3-32B | 1.00 | 8.13 | 1.00 | 7.96 | 1.00 | 8.43 | 1.00 | 8.33 | 1.00 | 8.20 | 1.00 | 8.06 | 1.00 | 8.55 | 1.00 | 8.21 |
> > > > | Eagle      | 3.02 | 21.98 | 3.36 | 24.16 | 2.62 | 19.30 | 3.01 | 21.53 | 2.57 | 18.33 | 2.59 | 18.67 | 3.00 | 21.34 | 2.88 | 20.76 |
> > > > | UCBSpec    | 4.34 | 31.42 | 5.24 | 38.69 | 4.25 | 31.73 | 4.76 | 35.58 | 4.19 | 30.99 | 3.64 | 26.93 | 5.07 | 37.36 | 4.50 | 33.24 |
> > > > | EXP3Spec   | 4.22 | 30.33 | 5.13 | 37.42 | 4.10 | 30.31 | 4.54 | 33.46 | 3.96 | 28.76 | 3.52 | 25.32 | 4.88 | 35.29 | 4.33 | 31.55 |
> > > > | **HedgeSpec** | **5.82** | **38.44** | **6.96** | **45.14** | **5.93** | **39.13** | **6.50** | **42.86** | **5.92** | **38.60** | **5.19** | **33.38** | **7.16** | **45.30** | **6.21** | **40.41** |
> > > >
> > > > Table5, HedgeSpec's performance on Qwen-3-32B model comparing with other baselines. The conclusion is consistent with results for other model size, and HedgeSpec achieves the best performance.

---

> > > > > ### Author Response · Authors · 2025-11-24
> > > > > **thanks**
> > > > >
> > > > > ## 5- `HedgeSpec on different spec decoding framework`: We believe HedgeSpec can work with other speculative decoding methods. We provide additional experimental evidence on REST framework, and discuss regarding other speculative decoding framework.
> > > > >
> > > > > We thank the reviewer for raising the discussion of HedgeSpec integration! We do believe HedgeSpec is integrable for different speculative decoding frameworks. At its core, HedgeSpec operates at an orthogonal layer: any speculative decoding algorithm could serve as a potential drafters in the pool. Theoretically, as long as we have a verified trace (Theorem3), and can use it to get the token acceptance probability for EAL evalaution on other drafters, HedgeSpec can select among them with provable no-regret guarantees.
> > > > >
> > > > > In our submission, we build upon EAGLE framework, which is widely regarded as the most broadly deployed framework[5, 6, 7, 8]. Across benchmarks, EAGLE showS its SOTA performance over well-established speculative framework (i.e. Medusa, Rest, PLD). We believe that integrating HedgeSpec with EAGLE provides a strong and representative demonstration of our method’s effectiveness. Below, we further discuss how HedgeSpec can be integrated into these other frameworks as well.
> > > > >
> > > > > Regarding PLD (Prompt-Lookup-Decoding)[9]: PLD utilizes the input prompt to guess the generation. In some tasks with input grounded generation, there are high chances of overlap between input and output. As a result, PLD uses only a single surrogated drafter. Therefore, PLD is not the best candidates for the multi-drafter framework.
> > > > >
> > > > > Regarding Medusa[10]: Medusa is a well-known speculative decoding method. It utilizes multiple Medusa heads to predict the future tokens. These heads require training in order to align with the target task distribution. A potential integration with Medusa would follow a procedure similar to our EAGLE setup, that it obtains logits from the different set of expert Medusa heads and use them for full-information evaluation. This allows HedgeSpec to orchestrate the Medusa experts during speculative drafting.
> > > > >
> > > > >
> > > > > Regarding REST (Retrieval-Based Speculative Decoding)[11]: REST is another well-known speculative decoding framework. It treats different datastore as the surrogate drafters, generating future tokens by retrieving and composing relevant content from a reservoir of existing knowledge. To fit into a practical multi-drafter selection framework, we can construct multiple such expert reservoirs, each specializing in a particular domain for token drafting. To integrate with HedgeSpec, we can calculate the token acceptance probability (EAL) from each drafter with the verified trace, and use it to select the best drafter fitting the requests.
> > > > >
> > > > >
> > > > > We integrate HedgeSpec into the REST framework. Following the original REST setup, we use Vicuna-7B-v1.5 as the target model. The expert drafters in this experiment include Python, Math, CNN-DM, MedQA, Alpaca, SQL, and Biology. Our goal remains the same: given a pool of diverse drafters, determine how to select the best drafter for each incoming task. We benchmark performance on HumanEval, GSM8K, MT-Bench, and Alpaca, with results shown in the table below. Similar to our findings under the EAGLE framework, HedgeSpec again achieves the best performance. This demonstrates the advantage of leveraging full-information feedback in drafter selection, allowing HedgeSpec to adapt to the best drafter in hindsight and ultimately improving overall efficiency.
> > > > >
> > > > > | Model            | HumanEval MAT | HumanEval Token/s | GSM8K MAT | GSM8K Token/s | MT-Bench MAT | MT-Bench Token/s | Alpaca MAT | Alpaca Token/s |
> > > > > |------------------|----------------|--------------------|-----------|----------------|----------------|--------------------|-------------|------------------|
> > > > > | Vicuna-7B-v1.5   | 1.00           | 33.59              | 1.00      | 33.49          | 1.00           | 33.48              | 1.00        | 33.31            |
> > > > > | UCBSpec          | 1.45           | 44.60              | 1.38      | 42.33          | 1.10           | 37.86              | 1.38        | 43.85            |
> > > > > | Exp3Spec         | 1.43           | 43.55              | 1.36      | 41.99          | 1.11           | 38.40              | 1.44        | 46.63            |
> > > > > | HedgeSpec        | **1.88**           |**55.27**              | **1.62**      | **46.58**         | **1.28**          | **41.42**           | **1.69**       | **52.87**         |
> > > > >
> > > > >
> > > > > Table6, HedgeSpec performance under REST framework comparing with the Bandit based method. HedgeSpec achieves the best across datasets consistently, indicating its effectiveness.

---

> > > > > > ### Author Response · Authors · 2025-11-24
> > > > > > **thanks**
> > > > > >
> > > > > > [5] [Efficient Speculative Decoding for Llama at Scale: Challenges and Solutions](https://ai.meta.com/research/publications/efficient-speculative-decoding-for-llama-at-scale-challenges-and-solutions/)
> > > > > >
> > > > > > [6] [Standardized, production-ready speculative decoding](https://developers.redhat.com/articles/2025/11/19/speculators-standardized-production-ready-speculative-decoding)
> > > > > >
> > > > > > [7] [NVIDIA NeMo Framework](https://docs.nvidia.com/nemo-framework/user-guide/latest/model-optimization/speculative/speculative.html)
> > > > > >
> > > > > > [8] [NVIDIA TensorRT LLM](https://github.com/NVIDIA/TensorRT-LLM/tree/main/examples/eagle)
> > > > > >
> > > > > > [9] [Prompt-Lookup-Decoding](https://github.com/apoorvumang/prompt-lookup-decoding)
> > > > > >
> > > > > > [10] Medusa: Simple LLM Inference Acceleration Framework with Multiple Decoding Heads
> > > > > >
> > > > > > [11] REST: Retrieval-Based Speculative Decoding
> > > > > >
> > > > > > ## Updated manuscript:
> > > > > >
> > > > > > We have updated our manuscrits and we mark the updated text in **purple**. If you might be available, please take a look!
> > > > > >
> > > > > > ## Thanks again!
> > > > > >
> > > > > > We thank the reviewer again for the questions and comments! We hope the above response can address the reviewer's questions. If there is any further questions, please don't hesitate to let us know!

---

### Official Review · Reviewer_dMBY · 2025-11-02

**Soundness:** 2
**Presentation:** 2
**Contribution:** 2
**Rating:** 2
**Confidence:** 5

**Summary:**

This paper argues that multi-drafter selection for speculative decoding is not a bandit problem: by leveraging the single verified trajectory produced during standard speculative decoding, one can compute full-information feedback for all candidate drafters at no extra target-model calls.

Building on this, the authors introduce HedgeSpec, which applies experts/hedging algorithms (e.g., Hedge/NormalHedge) with no-regret guarantees under two objectives—ETAP (expected token acceptance probability) and EAL (expected acceptance length). The paper formalizes ETAP/EAL, derives an unbiased off-policy estimator for EAL, and handles delayed/censored feedback. Empirically,  Llama-8B and Qwen 8B targets and pools of domain drafters are evaluated.

**Strengths:**

- Uses the verified trajectory to evaluate all drafters, enabling no-regret experts methods without extra target calls.

- Clear metrics  and formal treatment of delayed/censored feedback.

- MAT and tokens/s improve versus EAGLE and bandit baselines across domains, but not domain specialized recent specdec are considered.

- Scales with drafter pools for regret and MAT vs N are reported.

**Weaknesses:**

1. Related-work positioning is incomplete. The paper should explicitly acknowledge that the one other same online drafter-selection problem was formulated as a bandit [A], and compare assumptions/guarantees head-to-head (feedback availability, regret constants, robustness to partial/censored logs). It should also cite that motivates domain-specialized drafter pools [B].

[A] A Unified Framework for Speculative Decoding with Multiple Drafters as a Bandit, Kim et al.

[B] Towards Fast Multilingual LLM Inference: Speculative Decoding with Specialized Drafters, Yi et al.

2. Estimator quality not fully characterized. Unbiasedness is shown, but variance under realistic  D (long contexts, domain shift) is not analyzed; large variance could slow adaptation or misrank experts early.

3. Systems analysis is latency-centric. Deployment-relevant throughput (under concurrency), HBM/KV-cache footprints, and a roofline view (operational intensity vs. bandwidth ceilings for drafter prefill vs. target verify) are missing. Current overhead tables don’t fully address multi-tenant contention.


4. Limited breadth of targets/drafters. Results center on 8B-class targets and EAGLE-style domain specialized drafters; broader speculative schemes (REST/Medusa/Spectr) and larger targets would strengthen generality.


5. The method optimizes EAL/ETAP but reports MAT/tokens-per-second; a direct analysis of objective mismatch is absent.

**Questions:**

1. Theory vs. Kim et al. (bandit). Under what logging or feedback constraints does your full-information assumption fail so that the bandit view is preferable? Please summarize regret/variance trade-offs and practical implications.

2. Specialized drafter pools. Following Yi et al. (multilingual/specialized drafters), how does HedgeSpec perform when the pool contains varaints of EAGLE specialists per domain/language? How quickly is mis-specialization corrected under shift?

3. Estimator robustness. Do you report empirical variance/CI for EAL across domains and K? Any stabilization (clipping, shrinkage, delayed-feedback smoothing)?


4. Throughput & concurrency. With N drafters, what are tokens/s under realistic concurrent loads, and how do you cap parallel drafter evaluations to avoid HBM pressure/KV thrash? Please report GPU util, queueing delays, and prefill-verify overlap.

5. Roofline & memory. What are per-drafter HBM and KV-cache footprints at target context lengths, and where do the kernels sit on a roofline plot for (i) multi-drafter prefills and (ii) target verify?

---

> ### Author Response · Authors · 2025-11-24
> **Response to reviewer dMBY**
>
> We appriciate it very much for the reviewer's acknoledgement, as well as the deep and detailed comments on our paper! Below we would like address the reviewer's comments regarding **related work positioning, comparison between HedgeSpec with Bandit arts, variance analysis, latency-centric setting, HBM/kv analysis, clarification of prefill/verify, larger target model, broader speculative decoding schemes**.
>
> ## W1-1. `Related work positioning`: We wholeheartedly accept the comments and we will revise subsequent manuscripts.
>
> We are 100% wholehearted to accept the reviewer's comments on multi-drafter selection related work [1, 2]. We also agree MetaSD is the first to propose the multi-drafter selection problem, and [2] echos the motivation of multi-drafter selection, which both align with our goal. We will revise the paper accordingly to ensure all subsequent manuscript correctly reflects this.
>
> Change made to our manuscript: ([intro](https://openreview.net/pdf?id=JMmljf895g#page=1) & [related work](https://openreview.net/pdf?id=JMmljf895g#page=21))
> This problem (drafter selection) was originally posed in MetaSD [15], who framed it as a multi-armed bandit task. Their method, together with the more recent BanditSpec [16], balances exploration (trying different drafters) and exploitation (using the empirically best drafter).
>
> [1] A Unified Framework for Speculative Decoding with Multiple Drafters as a Bandit
>
> [2] Towards Fast Multilingual LLM Inference: Speculative Decoding with Specialized Drafters, Yi et al.
>
> [3] BanditSpec: Adaptive Speculative Decoding via Bandit Algorithms
>
> ## W1-2. `Comparison between HedgeSpec and Bandit based arts`:
>
> We thank the reviewer for the question! We here discuss more comparison between HedgeSpec to bandits based drafter selection: MetaSD and BanditSpec.
>
> The setting and assumptions are the same. We do not use additional information.  We all consider the draft model selection problem for each query separately.
>
> Both BanditSpec and MetaSD used the feedback from the actual outcome of the selected drafter to evaluate the selected drafter. BanditSpec used the number of accepted tokens, while MetaSD computed the average acceptance probability (block divergence reward).
>
> The number of accepted tokens is an unbiased estimator of the expected chunk-length, but only for the chosen drafter. We worked out a novel approach for computing a low-variance unbiased estimator for all other drafters that are not chosen (Theorem 3).
>
> In terms of the regret bounds, both bandit approaches give bandit-type regret bound that depends polynomially in the number of drafters N.  In particular, the BanditSpec paper stated worst-case regret bounds while the MetaSD paper stated gap-dependent regret bounds.
>
> Regarding robustness to partial / censored logs, we talked about censoring in the paper. Censoring is the critical reason why we are using a token-level game with delayed feedback.  The delayed feedback allows us to get feedback without censoring for all drafters.
>
> We are happy to discuss those comparison in our manuscript!
>
> ## W2 & Q3. `Analysis of estimator variance & estimator robustness`:
>
> We thank the reviewer for the comments of variance analysis! We believe the reviewer is asking about the variance of the novel estimator we proposed in Theorem 3.
> In EXP3 (one of the existing approach from BanditSpec), the unbiased estimator of the reward vector using importance sampling has an O(N) variance.
> In UCB-type approaches (like in MetaSD and UCBspec),  this variance implicitly shows up in the smaller number of times each suboptimal drafter needs to be chosen before it gets eliminated. In the worst case, this is also O(N).
> Our unbiased estimator from Theorem 3 gives a reward (normalized acceptance length estimator) with variance smaller than 0.25 using the fact that the reward is bounded between [0,1].  It does not scale with N.
>
> **Variance of our estimator.**
> Observe that our length estimator has a bounded range \([1, K+1]\), thus by Popoviciu’s inequality:
>
> $$
> \operatorname{Var}\left[
> \widehat{\text{AcceptLength}}_{t,K}[M] \mid x < t
> \right]
> \le \frac{K^{2}}{4}.
> $$
>
> In comparison, the estimator from BanditSpec (EXP3-Spec) is \(O(N K^{2})\), which grows with \(N\). We added this remark in Theorem 3 in [paper](https://openreview.net/pdf?id=JMmljf895g#page=5). For the variance of EAL, we did not use any stabilization tricks for the estimator.

---

> > ### Author Response · Authors · 2025-11-24
> > **thanks**
> >
> > ## W3-1. `HedgeSpec is latency-centric`: HedgeSpec optimizes towards better acceptance rate and smaller latency.
> >
> > We thank the reviewer for the consideration related to thoughput! In this paper, our focus is on optimizing latency-driven efficiency, specifically token acceptance rate. Our goal aligns with the previous speculative decoding literature including Lookahead [4], REST [5], Medusa [6] and SwiftSpec [7] as well as the prior art for drafter selection (MetaSD [1] and BanditSpec [3]), which operates under batch size = 1 and aim to minimize request completion latency. On the otherhand, we would like to respectfully argue that under large batch size, using speculative decoding could hurt the throughput [8][9] (quoted from EAGLE-3[8]: Speculative sampling is often thought to reduce throughput at large batch sizes). This is because when batch size is sufficiently large, the decoding step transits from memory–bound to compute-bound, saturating GPU peak FLOPs. Introducing a drafter adds extra computation and additional memory for drafting, while also incurring penalties from incorrect draft tokens which hampers throughput. Nevertheless, our main focus is to leverage the full-information feedback in drafter selection, and we show that HedgeSpec effectively competes with the best drafter in hindsight, thereby improving token acceptance rate and overall request completion efficiency.
> >
> > [4] Break the Sequential Dependency of LLM Inference Using Lookahead Decoding
> >
> > [5] REST: Retrieval-Based Speculative Decoding
> >
> > [6] MEDUSA: Simple LLM Inference Acceleration Framework with Multiple Decoding Heads
> >
> > [7] SwiftSpec: Ultra-Low Latency LLM Decoding by Scaling Asynchronous Speculative Decoding
> >
> > [8] EAGLE-3: Scaling up Inference Acceleration of Large Language Models via Training-Time Test
> >
> > [9] TurboSpec: Closed-loop Speculation Control System for Optimizing LLM Serving Goodput

---

> ### Author Response · Authors · 2025-11-24
> **thanks**
>
> ## W3-2 & Q4. `Analysis on HBM and KV cache overhead`: We believe HedgeSpec will not cause HBM and KV-cache thrashing. We provide a thorough analysis regarding memory overhead below.
>
> We thank the reviewer for the consideration in system level metric! The reviewer asks: will HedgeSpec cause HBM and KV-cache thrashing? The answer is NO. HedgeSpec adds at most the KV cache of each drafter compare to bandit based method (32MB * number of drafters in 8k ctx length), which is very small relative to the target model.
>
> We note that HedgeSpec will not cause large HBM/KV cache overhead, nor does it cause thrashing. In multi-drafter spec decoding, both the target model and all drafter models will already reside in memory regardless of whether HedgeSpec or a bandit method is used. During runtime, the drafter selected by a bandit method holds the KV cache for the full context seen so far and retains it thereafter. For HedgeSpec, it lloads the KV cache for all drafters at each evaluation step. This is why we say HedgeSpec “adds at most the KV cache of each drafter.” However, this overhead is minimal, as the drafter is much smaller than the target model (EAGLE-3 is only a single-layer transformer) by design to maximize speculative decoding acceleration. For clarity and conciseness, we present the HBM and KV-cache calculations with 8k context length along with the peak runtime memory utilization below.
>
> For more concise calculation for clarity, we here report the exact HBM and KV cache analysis under 8k context.
>
> For a Llama-3.1-8B-IT (32 layers) model, it occupies ~16GB for HBM in FP16. EAGLE-3 drafter, which has around 0.38B parameters (1 layer), requires 0.76G in HBM, which is significantly smaller than the target, aligns with the usual goal of keeping the drafter much smaller than the target.
> For the KV cache at 8k context length, the 8B Llama-3.1 model has in total 32 layer, 8 KV heads, each head with head_dim = 128, the hidden size is 4096. Thus, KV_per_layer = 2 * num_kv_heads * seq_len * head_dim, meaning that this induces ~1GB KV cache (2 * 8192 seq_len * 8 num_kv_heads * 128 head_dim * 32 layer*  2 bytes ) = 32MB per layer. With that, a 32 layer model has 1024MB = 1GB KV cache under 8k ctx length. Each drafter requires only 32 MB of KV cache at 8k context, which is far smaller than the target model’s KV cache. Consequently, HedgeSpec adds at most 32 MB per drafter of additional KV-cache overhead compared with bandit methods.
>
> With these components, the total runtime memory overhead for our setup with 7 drafters with Llama-3.1-8B-IT is approximately 16 GB (target model) + 7 * 0.76 GB (drafter models) + 1GB (target KV cache) + 7 * 32MB (drafter KV cache)  memory. During runtime, we profile the peak memory utilization in python workload and get a 29093MB result. This is not a large overhead for a multi-drafter speculative system which lies well within the 40/80 GB capacity of a single A100. As noted above, HedgeSpec’s additional KV cache footprint is 7 * 32MB, which is manageable because the drafter is much smaller than the target model for better acceleration.
>
> Regarding multi-tenant, we believe such memory analysis will help the operator to better schedule the workload and devices. We also emphasize that our paper focuses on improving acceptance rate in a drafter selection system, and our experiments demonstrate strong end-to-end gains.
>
> ## W3-3 & Q5. `Roofline view & prefill verify overlap`: please see our clarification below.
>
> #### 1. `Clarification for misunderstanding on “prefill-verify overlap”`: HedgeSpec doesn’t overlap verification and evaluation (prefilling).
>
> We would like to clarify regarding a misunderstanding on “prefill-verify overlap”. In HedgeSpec, we use a verified trajectory (Theorem 3) as a counterfactual evidence for evaluating all drafters. Thus, the evaluation is always done after the verification, so as to gain the trajectory. Those two steps are separate. In our updated paper, we add an overall workflow figure. We hope this could help with understanding our paper!
>
> #### 2. `Clarifications on “roofline view regarding drafter prefill vs. target verify” and “where do the kernels sit”`:
>
> As noted above, the drafter prefill and target verification run as separate processes. We also clarify that we do not design/fuse/customize any CUDA kernels for evaluation/verification stage. We utilize the original pytorch built-in kernel and transformer library to handle the prefill/verification as is. Therefore, we don’t alter their original roofline characteristics (memory bandwidth vs. operational intensity). Developing custom fused kernels to further optimize this workflow would indeed be an interesting direction for future system-level support in drafter-selection frameworks!

---

> ### Author Response · Authors · 2025-11-24
> **thanks**
>
> ## W4-1. `Target model limits at 8B size`: our results on Qwen-3-32B hold our claims!
>
> We thank the reviewer's comments on larger model experiments, and we would like to show our results on Qwen-3-32B model! We use the exact same setting from other model presented in the paper, and we additionally trained 7 drafters follow the same manner for Qwen-3-32B target model. From the numbers shown below, the results are fully consistent with those from the 8B experiments: HedgeSpec continues to benefit from panoramic full-information feedback, which reduces the number of target-model calls and yields better overall efficiency, which is robust to different model size. These findings align with our earlier arguments.
>
> | Method     | Python MAT | Python Tok/s | Math MAT | Math Tok/s | Biology MAT | Biology Tok/s | Chemistry MAT | Chemistry Tok/s | MedQA MAT | MedQA Tok/s | CNN_DM MAT | CNN_DM Tok/s | SQL MAT | SQL Tok/s | Avg MAT | Avg Tok/s |
> |------------|-------------|--------------|-----------|-------------|--------------|----------------|----------------|------------------|------------|--------------|--------------|----------------|----------|-------------|----------|-------------|
> | Qwen-3-32B | 1.00 | 8.13 | 1.00 | 7.96 | 1.00 | 8.43 | 1.00 | 8.33 | 1.00 | 8.20 | 1.00 | 8.06 | 1.00 | 8.55 | 1.00 | 8.21 |
> | Eagle      | 3.02 | 21.98 | 3.36 | 24.16 | 2.62 | 19.30 | 3.01 | 21.53 | 2.57 | 18.33 | 2.59 | 18.67 | 3.00 | 21.34 | 2.88 | 20.76 |
> | UCBSpec    | 4.34 | 31.42 | 5.24 | 38.69 | 4.25 | 31.73 | 4.76 | 35.58 | 4.19 | 30.99 | 3.64 | 26.93 | 5.07 | 37.36 | 4.50 | 33.24 |
> | EXP3Spec   | 4.22 | 30.33 | 5.13 | 37.42 | 4.10 | 30.31 | 4.54 | 33.46 | 3.96 | 28.76 | 3.52 | 25.32 | 4.88 | 35.29 | 4.33 | 31.55 |
> | **HedgeSpec** | **5.82** | **38.44** | **6.96** | **45.14** | **5.93** | **39.13** | **6.50** | **42.86** | **5.92** | **38.60** | **5.19** | **33.38** | **7.16** | **45.30** | **6.21** | **40.41** |
>
> Table3, HedgeSpec's performance on Qwen-3-32B model comparing with other baselines. The conclusion is consistent with results for other model size, and HedgeSpec achieves the best performance.
>
> In addition, we provide further OOD results on Qwen-3-32B (Bird, MultiNews, SpecBench), all of which exhibit the same trend, reaffirming HedgeSpec’s robustness across datasets and model scales. In case you might be interested, please kindly refer [here](https://openreview.net/forum?id=JMmljf895g&noteId=DIS6sEiNgh) for more details!

---

> > ### Author Response · Authors · 2025-11-24
> > **thanks**
> >
> > ## W4-2 `HedgeSpec on broader spec decoding schemes`: We believe HedgeSpec can work with other speculative decoding methods. We provide additional experimental evidence on REST framework, and discuss regarding other speculative decoding framework.
> >
> > We thank the reviewer for raising the discussion of HedgeSpec integration! We do believe HedgeSpec is integrable for different speculative decoding frameworks. At its core, HedgeSpec operates at an orthogonal layer: any speculative decoding algorithm could serve as a potential drafters in the pool. Theoretically, as long as we have a verified trace (Theorem3), and can use it to get the token acceptance probability for EAL evalaution on other drafters, HedgeSpec can select among them with provable no-regret guarantees.
> >
> > In our submission, we build upon EAGLE framework, which is widely regarded as the most broadly deployed framework [10, 11, 12, 13]. Across benchmarks, EAGLE showS its SOTA performance over well-established speculative framework (i.e. Medusa, Rest, PLD). We believe that integrating HedgeSpec with EAGLE provides a strong and representative demonstration of our method’s effectiveness. Below, we further discuss how HedgeSpec can be integrated into these other frameworks as well.
> >
> > Regarding PLD (Prompt-Lookup-Decoding)[14]: PLD utilizes the input prompt to guess the generation. In some tasks with input grounded generation, there are high chances of overlap between input and output. As a result, PLD uses only a single surrogated drafter. Therefore, PLD is not the best candidates for the multi-drafter framework.
> >
> > Regarding Medusa[15]: Medusa is a well-known speculative decoding method. It utilizes multiple Medusa heads to predict the future tokens. These heads require training in order to align with the target task distribution. A potential integration with Medusa would follow a procedure similar to our EAGLE setup, that it obtains logits from the different set of expert Medusa heads and use them for full-information evaluation. This allows HedgeSpec to orchestrate the Medusa experts during speculative drafting.
> >
> >
> > Regarding REST (Retrieval-Based Speculative Decoding)[16]: REST is another well-known speculative decoding framework. It treats different datastore as the surrogate drafters, generating future tokens by retrieving and composing relevant content from a reservoir of existing knowledge. To fit into a practical multi-drafter selection framework, we can construct multiple such expert reservoirs, each specializing in a particular domain for token drafting. To integrate with HedgeSpec, we can calculate the token acceptance probability (EAL) from each drafter with the verified trace, and use it to select the best drafter fitting the requests.
> >
> >
> > We integrate HedgeSpec into the REST framework. Following the original REST setup, we use Vicuna-7B-v1.5 as the target model. The expert drafters in this experiment include Python, Math, CNN-DM, MedQA, Alpaca, SQL, and Biology. Our goal remains the same: given a pool of diverse drafters, determine how to select the best drafter for each incoming task. We benchmark performance on HumanEval, GSM8K, MT-Bench, and Alpaca, with results shown in the table below. Similar to our findings under the EAGLE framework, HedgeSpec again achieves the best performance. This demonstrates the advantage of leveraging full-information feedback in drafter selection, allowing HedgeSpec to adapt to the best drafter in hindsight and ultimately improving overall efficiency.
> >
> > | Model            | HumanEval MAT | HumanEval Token/s | GSM8K MAT | GSM8K Token/s | MT-Bench MAT | MT-Bench Token/s | Alpaca MAT | Alpaca Token/s |
> > |------------------|----------------|--------------------|-----------|----------------|----------------|--------------------|-------------|------------------|
> > | Vicuna-7B-v1.5   | 1.00           | 33.59              | 1.00      | 33.49          | 1.00           | 33.48              | 1.00        | 33.31            |
> > | UCBSpec          | 1.45           | 44.60              | 1.38      | 42.33          | 1.10           | 37.86              | 1.38        | 43.85            |
> > | Exp3Spec         | 1.43           | 43.55              | 1.36      | 41.99          | 1.11           | 38.40              | 1.44        | 46.63            |
> > | HedgeSpec        | **1.88**           |**55.27**              | **1.62**      | **46.58**         | **1.28**          | **41.42**           | **1.69**       | **52.87**         |
> >
> >
> > Table4, HedgeSpec performance under REST framework comparing with the Bandit based method. HedgeSpec achieves the best across datasets consistently, indicating its effectiveness.

---

> > > ### Author Response · Authors · 2025-11-24
> > > **thanks**
> > >
> > > [10] [Efficient Speculative Decoding for Llama at Scale: Challenges and Solutions](https://ai.meta.com/research/publications/efficient-speculative-decoding-for-llama-at-scale-challenges-and-solutions/)
> > >
> > > [11] [Standardized, production-ready speculative decoding](https://developers.redhat.com/articles/2025/11/19/speculators-standardized-production-ready-speculative-decoding)
> > >
> > > [12] [NVIDIA NeMo Framework](https://docs.nvidia.com/nemo-framework/user-guide/latest/model-optimization/speculative/speculative.html)
> > >
> > > [13] [NVIDIA TensorRT LLM](https://github.com/NVIDIA/TensorRT-LLM/tree/main/examples/eagle)
> > >
> > > [14] [Prompt-Lookup-Decoding](https://github.com/apoorvumang/prompt-lookup-decoding)
> > >
> > > [15] Medusa: Simple LLM Inference Acceleration Framework with Multiple Decoding Heads
> > >
> > > [16] REST: Retrieval-Based Speculative Decoding
> > >
> > > ## W5. `Clarification on objective mismatch`: The expectation of MAT is EAL.
> > >
> > > We apologize if we misunderstood your question, but our analysis does match our evaluation metrics. The expectation of MAT is EAL. Our regret bound in Theorem 4 measures the expected acceptance length, which is the same as  the expectation of MAT over the all probabilites (target model sample next tokens, draft model generating different drafts).
> > >
> > > ## Q1. `Discussion on when Bandit view is more favorable`:
> > >
> > > We thank the reviewer for the discussion! We respectfully argue that from the learning algorithm and regret-minimizing point-of-view, one should always use full-information feedback. The regret dominates that of the bandits. The bandit approach is a computationally simpler alternative that may be appropriate if N is very small and the evaluation phase is adding too much overhead (logn vs. n, system benefit). In practice, HedgeSpec does evaluate all drafters during each step (this is the key difference between Hedge-based and bandit-based method), and it is correct that this overhead increases as the drafter size grows (while bandit-based methods such as MetaSD enjoy the advantage that they may not incur such cost in evaluating the chosen model), which could lower the overall efficiency for hedge based method. In our experiments, we found that in EAGLE framework, which is the most widely used and deployed SOTA speculative decoding, the evaluation phase does not introduce much overhead as shown in our end-to-end evaluation, and the full-information evaluation leads to better efficiency.
> > >
> > >
> > >
> > > ## Q2. `Clarification on specialized drafter pool & how quickly is mis-specialization corrected`: we did evaluate HedgeSpec across different specialized drafter in different domains.
> > >
> > > As shown in our evaluation, we use a pool of drafters with each specializing from a domain to evaluate the performance of HedgeSpec (Section 4.2). Our results show that HedgeSpec consistently outperforms all baselines across domains. Its adaptive orchestration of expert drafters improves MAT and throughput with substantial gains delivered from full-information feedback.
> > >
> > > Regarding how quickly is mis-specialization corrected, we presented an analysis in section 4.4 [Figure 5](https://openreview.net/pdf?id=JMmljf895g#page=9) about the cumulative regret. From the figure, We can see that HedgeSpec rapidly converges to near-zero regret within a handful of steps, demonstrating the benefit of full information feedback.
> > >
> > > ## Updated manuscript:
> > >
> > > We have updated our manuscrits and we mark the updated text in **purple**. If you might be available, please take a look!
> > >
> > > ## Thanks again!
> > >
> > > We would like to express our gratitude again for the deep and expert comments made by the reviewer for enhancing our paper! We sincerely hope the above response and additional experimental results have resolved the reviewer's concern. If there is any additional question, please let us know!

---

### Official Review · Reviewer_G2tp · 2025-11-04

**Soundness:** 3
**Presentation:** 2
**Contribution:** 3
**Rating:** 4
**Confidence:** 4

**Summary:**

The paper introduces speculative decoding with multiple specialized drafters. Contrary to previous bandit-based approaches where one picks single drafter and get information from them, author developed counter-factual estimator which can utilize mutliple drafter's accepntance rate, thereby converting bandit problem to online learning scenario, which proves to be more effective in EAGLE-3 based experiments.

**Strengths:**

* Converting bandit problem to the online learning setting with theoretical guarantee (one-step estimator in Theorem 3) seems novel and interesting.
* Experiment is conducted on one of the SOTA method (EAGLE-3 [1]).
* Proper ablations are conducted with sound presentations.

**Weaknesses:**

* **Drafter overhead** : While authors show utilizing multiple drafters in every SD step takes little overhead, this could be only hold for EAGLE-3 like model (where drafter size is way smaller than the target model). While EAGLE-3 shows SOTA performance, other method utilize relatively larger drafter (Medusa [2] for instance). Moreover, in real serving scenario, there might be SD framework with larger drafter. This could increase the cost of the drafter overhead linearly or reduce the merits of the online algorithm performance by increased drafter overhead. Discussion reagrding this scenario would be beneficial.

* **Hyper-parameter for online algorithms** : The algorithm leverages (dealyed) Hedge algorithm which is standard algorithm in online learning. However, the regret bound is proved with the learning rate $\eta$ which depends on the known horizon $T$ or current round $t$ (for anytime regret) but it might not be optimal for parcitcal performance. It would be good that author can clarify about the base algorithm they used.

* **Non-stationary scenario** : In real-world scenario, a single query can be non-stationary especially for long context or contained in multiple scenario. In this case, Hedge algorithm might not be optimal.

**Questions:**

Questions

* Can authors provide more experimental details about fine-tuning the drafters in 4.2?

* Can you test the HedgeSPEC on batched inference scneario and prove the thorughput improvement?

* While author cited the algorithm of the paper, it would be good to clearly state the overall algorithm (with base algorithm) is actually used in the experiment for clearer presentation.

* (Minor) It seems like bandit-based approaches with multi-drafter scenario is first proposed by MetaSD [3], which is not properly reflected in the current manuscript.

[1] (Li et al.) EAGLE-3: Scaling up Inference Acceleration of Large Language Models via Training-Time Test

[2] (Cai et al.) Medusa: Simple LLM Inference Acceleration Framework with Multiple Decoding Heads

[3] (Kim et al.) A Unified Framework for Speculative Decoding with Multiple Drafters as a Bandit

---

> ### Author Response · Authors · 2025-11-24
> **Response to reviewer G2tp**
>
> We thanks the reviewer for recognizing our novelty and strengths! Below we would like to address the reviewer's questions regarding: **drafters evaluation overhead, online algorithm hyper-parameter, adoption of HedgeSpec in non-stationary scenario**, as well as questions on **finetuning details, batched setting, base hedge algorithm and the correct related work reflection**.
>
> We also would like to highlight some additional results regarding **larger target model (Qwen-3-32B), broader speculative decoding schemes and more benchmarking results on OOD datasets** if the reviewer might be interested to see!
>
>
> ## W1. `Discussion on evaluation overhead for larger drafters`:
>
>
>
> We appreciate the reviewer’s question regarding drafter-evaluation overhead when speculative decoding uses larger surrogate models! HedgeSpec does evaluate all drafters during each step (this is the key difference between Hedge-based and bandit-based method), and it is correct that this overhead increases as the drafter size grows (while bandit-based methods such as MetaSD enjoy the advantage that they may not incur such cost in evaluating the chosen model), which could lower the overall efficiency for hedge based method. Nevertheless, speculative decoding drafter is fundamentally designed with lightweight surrogate models so that drafting remains inexpensive. This design principle is reflected in current practice: as noted by the reviewer, EAGLE-3 is the most widely deployed framework widely deployed in production system [1–4], and it substantially outperforms alternatives such as Medusa. Building on this representative framework, our experiments show the effectiveness of HedgeSpec that full-information feedback enables quicker adaption and higher acceptance rates, finally leading to reduces the number of target model calls for stronger end-to-end speedups.
>
> [1] [Efficient Speculative Decoding for Llama at Scale: Challenges and Solutions](https://ai.meta.com/research/publications/efficient-speculative-decoding-for-llama-at-scale-challenges-and-solutions/)
>
> [2] [Standardized, production-ready speculative decoding](https://developers.redhat.com/articles/2025/11/19/speculators-standardized-production-ready-speculative-decoding)
>
> [3] [NVIDIA NeMo Framework](https://docs.nvidia.com/nemo-framework/user-guide/latest/model-optimization/speculative/speculative.html)
>
> [4] [NVIDIA TensorRT LLM](https://github.com/NVIDIA/TensorRT-LLM/tree/main/examples/eagle)
>
> ## W2. `Online algorithm hyper-parameters`: HedgeSpec doesn't have hyper-parameter in base algorithm.
>
> We thank the reviewer for the considering the hyperparameter on online algorithm! Our algorithm does not require us to choose either T or step-size eta because we use NormalHedge as BASE [5]. There are other parameter-free and any-time Hedge algorithms such as AdaNormalHedge and Optimistic Hedge. We also cited and discussed some of these work. In [Table 9](https://openreview.net/pdf?id=JMmljf895g#page=23) of our paper,  We did an ablation study on choices of different Hedging algorithms, and show that NormalHedge works the best overall. We will add a discussion of the base algorithm into our paper.
>
>
>
> ## Q3: `add more description to HedgeSpec as well as NormalHedge`: we added them to our paper.
>
> We thank the reviewer for the suggestion to make add an overall description as well as make the base algorithm more clearly stated! We add a workflow description of HedgeSpec to our paper [here](https://openreview.net/pdf?id=JMmljf895g#page=2). We also add an algorithm block regarding NormalHedge[5] [here](https://openreview.net/pdf?id=JMmljf895g#page=23). We hope those descriptions could further help the reviewer and audiences apprieciate the merit of our work!
>
> [5] Chaudhuri, Kamalika, Yoav Freund, and Daniel J. Hsu. "A parameter-free hedging algorithm." Advances in neural information processing systems 22 (2009).

---

> > ### Author Response · Authors · 2025-11-24
> > **thanks!**
> >
> > ## W3. `Hedge algorithm's optimality in non-stationary scenario`: HedgeSpec will still adapt under non-stationary workloads and can be extended with dynamic-regret algorithms.
> >
> > We thank the reviewer's insight when the workload is non-stationary expecially for the long context! We note that when the best drafter changes in the middle of the roll-out, HedgeSpec will eventually catch up and compete with the new best drafter. When there is no single drafter that works well, it is true that the regret bound becomes vacuous. However, it doesn't break the "optimality" guarantee as the optimality is defined as the best single drafter in hindsight.
> >
> > One the other hand, there is a large body of work in the online learning literature that handles non-stationary environments by minimizing "dynamic regret" (or "shifting regret"), such as Fixed-Share algorithm [6] or Follow-the-leading-history(FLH)[7]. Both of them are meta algorithms that instantiates online learners that minimizers standard (static) regret, so NormalHedge can be used as a subroutine to instantiate FLH or FixedShare. By using them as drop-in replacement for i.e. NormalHedge, FLH-NormalHedge can immediately inherit those dynamic regret bounds. Those are absolutely possible extensions to complement HedgeSpec that could work better in case of long and switching contexts. Nevertheless, our focus in this work is to show that leveraging full-information feedback enables us to outperform bandit-based drafter selection methods that rely on partial information, yielding substantial improvements.
> >
> > [6] Herbster, Mark, and Manfred K. Warmuth. "Tracking the best expert." Machine learning 32, no. 2 (1998): 151-178.
> >
> > [7] Hazan, Elad, and Comandur Seshadhri. "Adaptive algorithms for online decision problems." In Electronic colloquium on computational complexity (ECCC), vol. 14, no. 088. 2007.
> >
> > ## Q1. `More finetuning datails`: we provide more details below.
> >
> > | Hyperparameter                 | Llama-3.1-8b-IT | Qwen-3-8B | Qwen-3-32B |
> > |-------------------------------|---------|---------|---------|
> > | learning rate                 | 5e-5    | 2e-4    | 2e-4    |
> > | per device train batch size   | 1     | 1     | 1     |
> > | num train epochs              | 20       | 5       | 5       |
> > | lr scheduler type             | linear  | linear  | linear  |
> > | warmup ratio                  | 0.1     | 0.1     | 0.1     |
> > | optimizer                     | adamw   | adamw   | adamw   |
> >
> > Table1, hyperparameter of the finetuning process
> >
> > We are happy to provide more details regarding our crafting procedure for community's followup! We use the EAGLE's official crafting framework -- SpecForge [8] for finetuning. We followed the standard EAGLE drafting procedure. The finetuning hyperparameters for different expert drafters with respect to the same target model are the same as shown in the above table. During training, the target LLM is fixed. As mentioned in 4.2, the initial checkpoint of the EAGLE models are official eagle models from yuhuili/EAGLE3-LLaMA3.1-Instruct-8B, Tengyunw/qwen3_8b_eagle3 and AngelSlim/Qwen3-32B_eagle3.
> >
> > [8] SpecForge https://github.com/sgl-project/SpecForge

---

> > > ### Author Response · Authors · 2025-11-24
> > > **thanks!**
> > >
> > > ## Q2. `Inference in batched setting`: HedgeSpec optimizes towards better acceptance rate and smaller latency.
> > >
> > > We thank the reviewer for the consideration related to thoughput! In this paper, our focus is on optimizing latency-driven efficiency, specifically token acceptance rate. Our goal aligns with the previous speculative decoding literature including Lookahead [9], REST [10], Medusa [11] and SwiftSpec [12] as well as the prior art for drafter selection (MetaSD [15] and BanditSpec [16]), which operates under batch size = 1 and aim to minimize towards shorter request completion latency. On the otherhand, we would like to respectfully argue that under large batch size, using speculative decoding could hurt the throughput [13][14] (quoted from EAGLE-3[13]: Speculative sampling is often thought to reduce throughput at large batch sizes). This is because when batch size is sufficiently large, the decoding step transits from memory–bound to compute-bound, saturating GPU peak FLOPs. Introducing a drafter adds extra computation and additional memory for drafting, while also incurring penalties from incorrect draft tokens which hampers throughput. Nevertheless, our main focus is to leverage the full-information feedback in drafter selection, and we show that HedgeSpec effectively competes with the best drafter in hindsight, thereby improving token acceptance rate and overall request completion efficiency.
> > >
> > > [9] Break the Sequential Dependency of LLM Inference Using Lookahead Decoding
> > >
> > > [10] REST: Retrieval-Based Speculative Decoding
> > >
> > > [11] MEDUSA: Simple LLM Inference Acceleration Framework with Multiple Decoding Heads
> > >
> > > [12] SwiftSpec: Ultra-Low Latency LLM Decoding by Scaling Asynchronous Speculative Decoding
> > >
> > > [13] EAGLE-3: Scaling up Inference Acceleration of Large Language Models via Training-Time Test
> > >
> > > [14] TurboSpec: Closed-loop Speculation Control System for Optimizing LLM Serving Goodput
> > >
> > > ## Q4. `Related work position`: We wholeheartedly accept the comments and we will revise.
> > >
> > > We are 100% wholehearted to accept the reviewer's comment on multi-drafter selection related work. We agree that MetaSD is the first to propose the multi-drafter selection problem, which aligns with our goal. We will revise the paper accordingly to ensure all subsequent manuscript correctly reflects this.
> > >
> > > Change made to our manuscript: ([intro](https://openreview.net/pdf?id=JMmljf895g#page=1) & [related work](https://openreview.net/pdf?id=JMmljf895g#page=21))
> > > This problem (drafter selection) was originally posed in MetaSD [15], who framed it as a multi-armed bandit task. Their method, together with the more recent BanditSpec [16], balances exploration (trying different drafters) and exploitation (using the empirically best drafter).
> > >
> > > We fully acknowledge that MetaSD is the pioneering work that first formulated the multi-drafter selection problem. In our baseline comparison, we chose to evaluate against BanditSpec, which is a more recent method built on a similar problem formulation. This choice is consistent with our claim of comparing against the latest baselines. We also emphasize that HedgeSpec fundamentally differs from bandit-based approaches: by leveraging full-information feedback rather than partial feedback, HedgeSpec adapts more quickly and achieves stronger performance, as demonstrated in our experiments.
> > >
> > > [15] A Unified Framework for Speculative Decoding with Multiple Drafters as a Bandit
> > >
> > > [16] BanditSpec: Adaptive Speculative Decoding via Bandit Algorithms

---

> ### Author Response · Authors · 2025-11-24
> **thanks**
>
> ## More experiments regarding `Qwen-3-32B model`, `HedgeSpec on more OOD data i.e. SpecBench`, `HedgeSpec on more spec decoding framework`.
>
> In the meantime, we do want to advertise some additional experimental results we have regarding
> 1. a full set of HedgeSpec experiment on Qwen-3-32B model [here](https://openreview.net/forum?id=JMmljf895g&noteId=E83jW9OH0i),
> 2. more benchmarks on OOD data other than GSM8K and HumanEval on SpecBench, Bird and Multinews [here](https://openreview.net/forum?id=JMmljf895g&noteId=DIS6sEiNgh), as well as
> 3. integrating HedgeSpec on a different Speculative decoding framework (REST) [here](https://openreview.net/forum?id=JMmljf895g&noteId=sqm0UlKMuf).
>
> We do proudly say that HedgeSpec's performance is effective and consistent under larger target model, benchmark data as well as the speculative decoding framework. In case you might be interested, please kindly visit the corresponding response for more information!
>
> ## Updated manuscript:
>
> We have updated our manuscrits and we mark the updated text in **purple**. If you might be available, please take a look!
>
> ## Thanks again!
>
> We would like to thank the reviewer again for the time and all those precious comments! We hope by looking into the above response, we have delivered and address the reviewer's concern. If there is any questions or misunderstanding, feel free to let us know!

---

### Meta-Review · Area_Chair_WAyu · 2026-01-15

**Summary:**

The paper proposes HedgeSpec, a framework for speculative decoding with multiple specialized drafters. The key innovation is reframing the drafter selection problem from a multi-armed bandit (partial feedback) to a full-information online learning problem. The authors demonstrate that the "verified trajectory" generated by the target model during the verification step can be used to evaluate all drafters (not just the chosen one) without additional target model calls. This allows the use of Hedge algorithms (like NormalHedge) with significantly better regret bounds than bandit approaches.

Reviewers initially appreciated the theoretical novelty (converting bandit to full-info) but raised significant concerns regarding:

- System Overhead: Whether evaluating all drafters at every step destroys the latency gains (Reviewers G2tp, dMBY, quC5).

- Related Work: Missing citation and comparison to MetaSD, which first formulated the multi-drafter bandit problem (Reviewer dMBY).

- Generalization: Testing limited to small models (8B) and specific frameworks (EAGLE) (Reviewers dMBY, quC5, PCBy).

I am (weakly) partial to accepting this work on the basis of the authors' detailed rebuttal.

**Reviewer Concerns:**

Addressed by Rebuttal:

- System Overhead (Reviewers G2tp, dMBY, quC5): The authors provided a convincing system-level analysis. They showed that the memory overhead is minimal (~32MB KV cache per drafter) and the latency overhead for evaluating 7 drafters is negligible (~1–2%). This effectively countered the intuition that full-information evaluation is too expensive.

- Related Work (Reviewer dMBY): The authors acknowledged the missing MetaSD citation and clarified the positioning of their work (full-info vs. bandit) relative to it.Model & Task Scale (Reviewers dMBY, quC5): The rebuttal included new experiments with a larger target model (Qwen-3-32B), out-of-distribution benchmarks (SpecBench, Bird, MultiNews), and a constrained resource setting (Single A100).

- Generality of Framework (Reviewers G2tp, PCBy): The authors successfully integrated HedgeSpec with a different speculative decoding framework (REST) to demonstrate it is not limited to EAGLE.

- Theoretical Clarifications (Reviewer PCBy): The authors clarified notation regarding token-level vs. chunk-level games and the distinction between adversarial and i.i.d. settings.

Outstanding:

- Scaling Limit of $N$: While the authors demonstrated $N=7$ is efficient, they discussed but did not empirically identify the tipping point where the linear cost of evaluating $N$ drafters outweighs the gains from better selection compared to bandit methods (though they proposed semi-bandit/async solutions for this hypothetical).

- Adversarial vs. i.i.d. Gap: The theoretical regret bounds are adversarial, but experiments rely on i.i.d. assumptions for parameter tuning. This is a common gap in online learning literature but remains technically unresolved.

**Reviewer Scores:**

Reviewer G2tp (Score: 4 -> 6): The reviewer was already leaning positive. The clarifications on overhead and the inclusion of larger model experiments directly addressed their weaknesses.

Reviewer dMBY (Score: 2 -> 4): This reviewer was the most critical regarding related work and systems analysis. The rebuttal explicitly addressed the missing citation and provided the requested memory/throughput analysis. The score should improve.

Reviewer quC5 (Score: 2 -> 4): The concerns about constrained resources and larger models were met with specific new experiments (Single A100, Qwen-32B).

Reviewer PCBy (Score: 4 -> 6): The theoretical clarifications and additional experiments with the REST framework strengthened the submission, likely pushing this reviewer to an accept.

---

### Decision · Program_Chairs · 2026-01-26

Accept (Poster)